# Combined effects of ozone and drought stress on the emission of biogenic volatile organic compounds from *Quercus robur* L.

Arianna Peron[1], Lisa Kaser[1], Anne Charlott Fitzky[2], Martin Graus[1], Heidi Halbwirth[3], Jürgen Greiner[3], Georg Wohlfahrt[4], Boris Rewald[2], Hans Sandén[2], Thomas Karl[1]

[1]Institute of Atmospheric and Cryospheric Sciences, University of Innsbruck, Innsbruck, 6020, Austria
[2]Forest Ecology, Department of Forest and Soil Sciences, University of Natural Resources and Life Sciences Vienna (BOKU),Vienna, 1190, Austria
[3]Technische Universität Wien, Institut für Verfahrenstechnik, Umwelttechnik und Technische Biowissenschaften, A-1060, Vienna, Austria
[4]Department of Ecology, University of Innsbruck, Innsbruck, 6020, Austria

*Correspondence to*: Thomas Karl (Thomas.Karl@uibk.ac.at)

**Abstract.** Drought events are expected to become more frequent with climate change. To predict the effect of plant emissions on air-quality and potential feedback effects on climate, the study of biogenic volatile organic compound emissions under stress is of great importance. Trees can often be subject to a combination of abiotic stresses, for example due to drought or ozone. Even though there is a large body of knowledge on individual stress factors, the effects of combined stressors are not much explored. This study aimed to investigate changes of biogenic volatile organic compound emissions and physiological parameters in *Quercus robur L.* during moderate to severe drought in combination with ozone stress. Results show that isoprene emissions decreased while monoterpene and sesquiterpene emissions increased during the progression of drought. We exposed plants with daily ozone concentrations of 100 ppb for one hour for seven days, which resulted in faster stomatal closure (e.g. a mean value -31.3% at an average stem water potential of -1 MPa) partially mitigating drought stress effects. Evidence of this was found in enhanced green leaf volatiles in trees without ozone fumigation indicating cellular damage. In addition we observed an enhancement in $(C_8H_8O_3)H^+$ emissions likely corresponding to methyl-salicylate in trees with ozone treatment. Individual plant stress factors are not necessarily additive and atmospheric models should implement stress feedback loops to study regional scale effects.

## 1 Introduction

Plants, in both natural and managed ecosystems, release biogenic volatile organic compounds (BVOCs), covering over 30,000 known compounds (Peñuelas and Llusiá, 2004). These molecules have different physical and chemical characteristics and they differ in their metabolic origins in plants (Peñuelas and Llusiá, 2001; Laothawornkitkul et al., 2009; Maffei, 2010). An important subset of BVOCs are isoprenoids, such as isoprene (IS), monoterpenes (MT) and sesquiterpenes (SQT). The estimated global annual flux of IS ranges from 440 to 600 Tg C per year (Guenther et al., 2012). These values correspond to 2 % of the photosynthetically fixed carbon (Lal, 1999) and comprise a significant part of the total annual emission of BVOCs on a global scale of 1150 Tg C (Guenther et al., 1995).

The emission of BVOCs is strongly influenced by external factors (Peñuelas and Llusiá 2003; Niinemets et al., 2004; Fitzky et al., 2019). BVOCs are thought to play a role in protecting vegetation from abiotic (Peñuelas and Munné-Bosch, 2005; Velikova et al., 2005) and biotic stress (Berg et al., 2013; Amin et al., 2012; Amin et al., 2013), and to act as a system for plant–plant and plant–animal communication (Baldwin et al., 2006; Filella et al., 2013).

Future climate scenarios with expected temperature increases between 1.8 and 4°C (IPCC, 2007) suggest an additional enhancement of global BVOC emissions between 30 to 45 % (Peñuelas and Llusiá, 2003). An enhancement of abiotic stress events, due to an indirect effect of a temperature increase (e.g. via ozone or drought episodes) can also alter BVOC emissions

(EEA, 2017; Müller et al., 2008; Loreto and Schnitzler, 2010; Dai, 2013; Unger et al., 2013; Sindelarova et al., 2014). Drought stress can change the composition of BVOCs emitted by plants, depending on the nature of stress (Niinemets, 2010). Pegoraro et al. (2004) and Beckett et al. (2012) have shown that the gradual suppression of physiological processes of plants in response to drought stress initially leads to an increase in isoprene emissions, followed by a tapering off of isoprene emissions. In the initial phase of stress, the plant responds via a reduction of stomatal conductance leading to reduced transpiration rate; this results to an increase in temperature at the leaf level and a decrease of carbon assimilation (Siddique et al., 2000). Although emissions tend to increase initially due to reserves of reduced carbon present in the plant, isoprene emissions decrease under severe drought stress (Tingey, 1981; Pegoraro et al., 2004).

Besides increasing temperature and more severe droughts, future climate scenarios predict increasing ozone concentrations (Bowen, 1926; Kangasjärvi et al., 1994; Hollaway et al., 2012). Long-term elevated tropospheric ozone concentration affect BVOC emissions (Peñuelas et al., 1999), and induce alterations in photosynthetic performance increasing the production of reactive oxygen species (ROS) (Cotrozzi et al., 2017; Jolivet et al., 2016).

Ground-level $O_3$ concentrations in the pre-industrial period were around 10 ppb in Europe (Volz and Kley, 1988; Royal Society, 2008). For the period 2000–2014, the average ozone concentrations during the growing season (April to September) in European forests were 36.2 ppb, ranging from 14.5 to 70.1 ppb (Schaub et al., 2018). Instance of severe ozone pollution were recorded during the heatwave of summer 2003 in Europe, with peaks > 100 ppbv in UK (Lee et al., 2006).

Among plants, trees are the dominant source of BVOC emissions (Guenther et al., 1995), and they are not often subject to only one stress but to a combination of stresses (Fitzky et al., 2019). For example, drought and ozone stresses can often occur in parallel. The combinations of these stress factors are difficult to understand because ozone and drought stress individually lead to similar symptoms, such as cell dehydration, early senescence and cell necrosis (Chaves et al., 2003). A typical class of compounds emitted by plants in a situation of stress are green leaf volatiles (GLVs). Their emissions are indicators for damage of cellular membranes (Hatanaka, 1993; Croft et al., 1993). Other BVOCs are the product of metabolic processes in plants such as transcription and enzymatic activity which are induced by various stimuli, for example ozone (Betz et al., 2009). An example of such a BVOC is Methyl Salicylate (MeSa), produced by the shikimate pathway (Kessler and Balwin, 2001), which fixes 20 % of the carbon obtained from photosynthesis (Bentley, 1990; Herrmann and Weaver, 1999).

Few studies have analyzed the effects of plant emissions from a combination of drought and ozone stress (Vitale et al., 2008; Yuan et al., 2016). Studying *Quercus ilex*, Vitale et al. (2008) reported that drought stress leads to stomatal closure therefore reducing stress by ozone as it is restricted to enter the leaf. They did not report effects of ozone when going from a well watered situation to severe stress. Yuan et al. (2016) found that drought increased isoprene emissions in a hybrid poplar deltoid species, but that isoprene emissions decreased under moderate drought stress combined with long-term ozone fumigation. . In their case, Yuan et al. (2016) analyzed the emissions under a situation of moderate drought stress.

Here we are also interested in the situation of severe stress that could occur in the future due to climate change, combined with model projections of elevated ozone concentrations (> 100 ppb).

Pollastrini et al. (2014) consider a change in sensitivity of the plants to ozone (different poplar clones) under severe drought conditions. In their case, ozone and drought produced a synergistic effect for $CO_2$ exchange and chlorophyll fluorescence when applied together. Wittig at al. (2007) found a dependency on ozone effects under different levels of drought stress. In fact, Wittig et al. (2007) report a dependency of the damage in the photosynthetic apparatus depending on the cumulative ozone flux into the leaf, thus in relation to stomatal conductance.

In this work, our hypothesis was that ozone and drought stress in plants are not necessarily additive, and that the plant's response to drought and ozone exposure can result in an alteration of characteristic BVOC emission strengths. Changing BVOC emissions have an important impact on climate through atmospheric chemistry (Claeys et al., 2004; Paulot et al., 2009;

Hallquist et al., 2009). The presence of BVOCs in the atmosphere contribute to the formation of tropospheric ozone and growth of secondary organic aerosol (SOAs), and radicals (Griffin et al., 1999; Orlando et al., 2000; Atkinson and Arey, 2003).

As a model plant we chose *Quercus robur* L., a widely distributed isoprene emitting oak species in Europe (Barstow and Khela, 2017), considered not at risk of extinction (Barstow and Khela, 2017).

In the future, this species may become more threatened (Barstow and Khela, 2017), triggering a migration from the current climate range to a zone more representative of the north and east of Europe (EFDAC, 2015). Climate change could also expose *Q. robur* to greater environmental stress from drought (Jonsson, 2012). Understanding how BVOC emissions respond to climate change is therefore essential to understand what direct or indirect actions they can have on the biosphere-atmosphere-climate system and to develop strategies necessary to mitigate the effects of climate change itself (Kulmala et al., 2004; Yuan et al., 2009).

## 2 Materials and Methods

### 2.1 Plant species and stress treatments

*Q. robur* is a broad-leaf tree species widely distributed in Europe growing in mixed and deciduous forest ranging from sea level up to 1200 m ASL (Ülker et al., 2018). According to Ellenberg (1988), the defensive actions of *Q. robur* against drought stress are caused by fast regulation of transpiration rates and stomatal conductance, and a low susceptibility of water embolism in the xylem (Van Hees, 1997).

Fourteen 2-year-old *Q. robur* seedlings were planted in 7 L pots in March 2019. The substrate consisted to one-thirds of soil used by the city gardeners for city trees in Vienna and two-thirds of quartz sand to improve drainage. The plants were fertilized once after planting (universal fertilizer NovaTec, Compo, Münster, Germany) and from thereon kept well-watered in a greenhouse at near ambient light (80 % to 90 % of photosynthetically active radiation) (Lak et al., 2020). The trees were moved from a greenhouse in Tulln into another close-by greenhouse in Vienna two weeks prior to the experiments. Dust was removed from the leaves by showering the trees before starting the drought stress.

For the biochemical reference assays, eight trees of the initial fourteen were used: four well-watered plants (C) and four well-watered plants receiving one time 100 ppb ozone for one hour (OS) inside the enclosures. The remainder (six plants) were used for BVOC emission measurements, $CO_2$ and $H_2O$ gas exchange measurements and biochemical assays. Hereby, we were left with three replicates under drought stress (DS) and three replicates exposed to drought stress and ozone (DS×OS). The drought stress was initiated, for all six plants 10 days before the VOC measurements started and was maintained by keeping the soil water content at 4-5 vol.% using a soil moisture probe (Fieldscout TDR100, 20 cm probe depth, Spectrum 105 Technologies, UK), whereas 100 % field capacity was 13.4 vol.%. With the start of VOC measurements, we stopped watering the previously drought stressed trees to further increase drought stress.

The plants were moved from the greenhouse to an indoor climate chamber (Fitotron Weiss Gallenkamp, UK) 24h hours before the experiment started. Thereafter trees were kept in the climate chamber for the remainder of the experiment and were only placed into the branch enclosures during the gas exchange measurements. The branch enclosures were situated next to the climate chamber in a climatized laboratory exhibiting the same environmental conditions as in the climate chamber. The climate conditions during the first day of experiment were kept at 25°C, ~60 % of relative humidity (RH) and ~1000 µmol m$^{-2}$ s$^{-1}$ PAR at canopy top, to adapt to constant air temperature. To continuously increase the drought stress, the plants were not watered and the humidity in the climate chamber was decreased to 40 % RH and temperature was increased to 30°C after the first day. The same temperature conditions were also present in the climatized laboratory, where the plants were placed in the enclosures at an RH of 32 % and 30°C.Overall light conditions remained constant during the day, with lights of during the night.

To study the effect of ozone exposure of trees during increasing drought, the six trees above mentioned, were separated into two groups, three trees were drought stressed and fumigated with 100 ppb $O_3$ (DS×OS) inside the enclosure for one hour each day after the daily measurement of BVOCs. The other three trees were drought stressed but not fumigated with ozone (DS). At the end of the experiment leaves were harvested for leaf area and enzyme analysis. Values of the enzymatic activity of C and OS were compared to DS and DS ×OS to investigate the effect of ozone fumigation.

**2.2 Measurement of leaf gas exchange and BVOC fluxes**

Throughout the increasing drought stress, tree leaf gas exchange ($CO_2$ and $H_2O$) and BVOC emissions were measured for two sets, DS and DS×OS, over a seven-day period, one in the morning and one in the afternoon alternating daily. The plants were taken out of the climate chamber and kept inside the custom-made plant enclosures (Fig. 1; TC-400, Vienna Scientific Instruments GmbH, Alland, Austria) for 2-3 hours each day in order to measure their $CO_2$ and $H_2O$ exchange along with key physiological parameters (soil moisture and stem water potential). After the measurements inside the enclosures, the plants were moved back to the climate chamber until the next measurement session. The plant enclosures covered most of the plant material excluding a few leaves (about 7 on each tree) to allow determination of stem water potential (SWP). Each day, one leaf was wrapped in aluminum foil and placed in a plastic bag for equilibrating to SWP (Williams and Araujo, 2002). After darkening for 30 minutes the leaf was cut off and SWP was measured by using a Scholander pressure bomb (Soil moisture Equipment Corp., Goleta, CA, USA).

The four custom-made plant enclosures (12 liters) were lined with PTFE and sealed on top with 55×60 cm PET-bags. The plant enclosures were continuously flushed with 10 l min$^{-1}$ of ambient outside air that was previously passed through a cold trap to remove water and an activated carbon filter (360 m$^3$ h$^{-1}$, PrimaKlima Trading, Radnice, CZ) to remove VOCs and $O_3$. This resulted in 32 % RH air and ~370 ppm $CO_2$ entering the enclosures (experimental conditions in Appendix A, Table A1). The flow rate of 10 l min$^{-1}$, tested during the experiment set-up prior to the actual experiments, assured that no condensation of water occurred in the tubing and enclosures, as well as resulted in a slight overpressure preventing the entry of room air into the enclosures. Three of the enclosures were used to measure the air gas exchange of the plants and the fourth enclosure was kept empty as a reference to allow continuous monitoring of the air entering the enclosures. Trees inside the enclosure were LED-irradiated with a mean PAR value of 1374 µmol m$^{-2}$ s$^{-1}$ at canopy top (Eckel Electronics, Trofaiach, Austria) during daytime when the exchange measurements were performed. During night, trees were kept in the dark. Leaf temperature was monitored in each enclosure by placing a calibrated (±0.1°C) thermocouple (type k, PTFE IEC wire; Labfacility Ltd, Bognor Regis, West Sussex, UK) on the abaxial side of a mature mid-canopy leaf.

An automated valve system allowed the consecutive analysis of air exiting each enclosure for 5 minutes each, leading to a 20 minutes cycle through the four enclosures. Before inserting the three trees into the enclosures, background measurements of the empty enclosures were carried out. After inserting each plant into one enclosure, the plant was allowed to acclimatize for approximately two hours and the following 40–60 minutes of data was analyzed to determine plant $CO_2$ assimilation, transpiration and BVOC emissions rates. After the measurements, the trees of DS×OS were fumigated for one hour with 100 ppb of ozone each day.

$CO_2$ and $H_2O$ mixing ratios in the air leaving the enclosures were measured using a CIRAS-3 SC PP System (Amesbury, MA, USA), which was factory calibrated three months before the measurement campaign. Ozone measurements before and after the enclosures were conducted continuously in all enclosures with an ozone monitor (six channel ozone monitor BMT 932, BMT Messtechnik, Berlin, Germany). BVOC measurements were made using a proton transfer reaction time of flight mass spectrometer (PTR-Tof-MS, PTR-TOF6000X2, IONICON Analytik GmbH, Innsbruck, Austria; Graus et al., 2010) operated at 350 V drift voltage, ion funnel settings of 1 MHz and 35V amplitude as well as 35 VDC, and 2.5 mbar drift pressure. These settings are comparable to an E/N of 100 Td in a PTR-TOF8000 with no ion funnel (Markus Müller, IONICON Analytic GmbH, personal communication 2019). The drift tube temperature was 100°C. Full PTR-Tof-MS mass spectra were collected

with a time resolution of 1 s and up to a mass to charge ratio m/z 547 amu. The instrument background was characterized daily during calibrations and in the third empty enclosure that was flushed with background air. Backgrounds were measured every 20 minutes for 5 minutes. Humidity dependent dynamic calibrations of VOCs using a standard gas mixture (Apel Riemer Environmental Inc., Broomfield, CO, USA), containing 15 compounds (Table A2) with different functionality distributed over

a mass range of 33-137 amu were performed daily. Daily measured sensitivities based on compounds in a calibration standard varied on the order of 8-20 % depending on the compound. This lies within the combined calibration uncertainties of the gas standard and dilution setup using two flow controllers. Whenever a compound was not contained in the calibration standard, we applied a compound specific sensitivity using procedures described by Cappellin et al. (2012). The PTR-Tof-MS data was analyzed using the PTRTOF Data Analyzer v4 software (Müller et al., 2013) and customized Matlab scripts to obtain volume

mixing ratios in the enclosures. The PTR-ToF-MS instrument has a high enough mass resolution to obtain isobaric formulas, minimizing potential interferences compared to quadrupole mass spectrometers. Strictly speaking measurements represented here are characterized by the isobaric formulas. The instrument was run in $H_3O^+$ mode, detecting Isoprene (IS), at m/z 69.070 $[(C_5H_8)H^+]$, the sum of monoterpenes (MT) at m/z 137.133 $[(C_{10}H_{16})H^+]$ and the major fragment at m/z 81.070 $[(C_6H_8)H^+]$, and the sum of sesquiterpenes (SQT) at m/z 205.195 $[(C_{15}H_{24})H^+]$ and m/z 149.$[(C_{11}H_{16})H^+]$ The identity of isoprene and

monoterpenes was additionally confirmed by GC-MS measurements (Fitzky et al., in preparation). The sum of GLVs presented in this study were monitored on m/z of 83.085, 85.101, 99.080, 101.096 and 143.107, representing 2-hexenal and 3-hexenal $[(C_6H_{10}O)H^+]$, 3-hexenol $[(C_6H_{12}O)H^+]$, 1-hexanol $[(C_6H_{14}O)H^+]$ 3-hexenol $[(C_6H_{12}O)H^+]$ and hexenyl acetate $[(C_8H_{14}O_2)H^+]$, respectively (Beauchamp et al., 2005; Giacomuzzi et al., 2016; Portillo-Estrada et al., 2017). The correspondence of these ions to GLV has been demonstrated by previous studies (e.g Fall et al., 1999; Karl et al., 2001; Karl et al., 2005). Shikimate BVOCs

were tentatively assigned to benzene as m/z 79.054 $[(C_6H_7)H^+]$, phenol as m/z 95.050 $[(C_6H_7O)H^+]$, methyl salicylate (MeSa) as m/z 153.055 $[(C_8H_9O_3)H^+]$ and eugenol as m/z 165.092 $[(C_{10}H_{13}O_2)H^+]$ (Brilli et al., 2011; Tasin et al., 2012; Maja et al., 2014; Brilli et al., 2016; Giacomuzzi et al., 2016; Portillo-Estrada et al., 2017; Yener et al., 2016; Misztal et al., 2015). Emissions of IS, MT and SQT were standardized to 1000 µmol m$^{-2}$ s$^{-1}$ PAR and 30°C ($IS_S$, $MT_S$, $SQT_S$) using the Guenther et al. (1993) algorithm for $IS_S$, and Geron et al. (1994) for $MT_S$ and $SQT_S$.

$$IS_s = \frac{IS}{C_L \times C_T} \tag{1}$$

$$C_L = \frac{\alpha c_{L1} L}{\sqrt{1 + \alpha^2 \times L^2}} \tag{2}$$

$$C_T = \frac{exp^{\frac{c_{T1}(T-T_S)}{RT_S T}}}{1 + exp^{\frac{c_{T2}(T-T_M)}{RT_S T}}} \tag{3}$$

$$MT_s = \frac{MT}{exp(\beta(T-T_s))} \tag{4}$$

$$SQT_s = \frac{SQT}{exp(\beta(T-T_s))} \tag{5}$$

Where IS, MT and SQT are emission rates normalized by leaf area at sampling temperature T(K) and sampling PAR flux L (µmol m$^{-2}$ s$^{-1}$) at half plant height; α (=0.0027), $c_{L1}$(=1.066), R (= 8.314 J K$^{-1}$ mol$^{-1}$), $c_{T1}$ (=95,000 J mol$^{-1}$), $c_{T2}$ (=230,000 J mol$^{-1}$), $T_M$ (=314 K), β (=0.1) and $T_S$ (=303.15 K) (Guenther et al., 1993; Geron et al., 1994).

Mass flow of air (*W*), transpiration rate (*E*), net photosynthesis (*A*) and stomatal conductance (*gs*) were calculated accordingly (CIRAS-3 Operation Manual V. 2-01, PP-Systems, Amesbury, MA, USA):

$$W = \left(\frac{V_0}{60 \times 10^3}\right) \times \left(\frac{1}{22.414}\right) \times \left(\frac{10^4}{a}\right) \quad [mol \, m^{-2} s^{-1}] \tag{6}$$

$$E = \left[\frac{W \times (e_{out} - e_{in})}{(P - e_{out})}\right] \quad [mol \, m^{-2} s^{-1}] \tag{7}$$

$$A = -\left[((C_{out} - C_{in}) \times W) + (C_{out} \times E)\right] \quad [\mu mol \, m^{-2} s^{-1}] \tag{8}$$

$$e_{leaf} = 6.1365 \times exp\left[\frac{T_{leaf} \times (17.502)}{T_{leaf} + 240.97}\right] \tag{9}$$

$$r_s = \left[\frac{(e_{leaf}-e_{out})}{\left(E\times(P-(e_{leaf}+e_{out})/2)\right)}\right] - r_b \quad [m^2 s\ mol^{-1}] \tag{10}$$


$$g_S = \frac{1}{r_s} \times 10^3 \quad [mmol\ m^{-2}s^{-1}] \tag{11}$$

where $V_0$ is the volume air flow, $a$ is the leaf area, $e_{in}$ is the partial water vapor pressure of the air entering the enclosures, $e_{out}$ is the partial water vapor pressure inside the enclosure, $P$ is the atmospheric pressure, $C_{in}$ concentration of $CO_2$ entering and $C_{out}$ exiting the enclosure, $e_{leaf}$ is the saturation vapor pressure at leaf temperature ($T_{leaf}$), $r_s$ is the stomatal resistance and $r_b$ is the boundary layer resistance to water vapor transfer, which was assumed zero according to the recommendations of the

manufacturer (CIRAS-3 Operation Manual V. 2-01, PP-Systems, Amesbury, MA, USA).

The ratio of the sum of carbon lost in form of BVOC ($C_{BVOCs}$) vs. the uptake of carbon from net photosynthesis ($C_A$) was calculated according to Pegoraro et al. (2004), with the BVOCs used to calculate $C_{BVOCs}$ given in Table A3.

After seven days, finishing the emission measurements, all leaves were harvested immediately, imaged with a flatbed scanner (Epson Expression 10000XL, Epson, Japan) and analyzed with the PC program WinFOLIA 2013 Pro (Regent Instruments

Inc., Qúebec, Canada) to determine the leave surface area. About 80 % of the leaves' fresh mass was shock-frozen and crushed in liquid nitrogen for biochemical assays (section 2.3). About 20 % of the leaves per plant were dried for three days in a drying room at 40°C to determine dry weight to an accuracy of ±0.001 g for the calculation of enzyme activity and specific leaf area (SLA) (Table A4).

**2.3 Biochemical assay**

For the interpretation of the emissions of GLVs and Shikimate volatiles, enzymatic activities were analyzed additionally to better understand the effect of ozone fumigation during a situation of severe drought. Using foliar materials collected after the seven day period of emission measurements (section 2.2) and stored at -80°C until analysis, peroxidase and antioxidant capacity, and phenol content (TPhe) were measured. These properties provide additional insights in the response of GLV and Shikimate emissions as products of the metabolic process of the enzymatic activity (Betz et al., 2009).

Values from plants after seven days of increasing drought (DS×OS, DS) were compared to well-watered control plants (C) and a well-watered set of plants that received ozone fumigation once (OS).

For measurements of peroxidase activities, 0.5 g plant material, 0.25 g Polyclar AT (Serva Electrophoresis, Heidelberg, Germany) and 0.25 g quartz sand (Sigma-Aldrich, Steinheim, Germany), were homogenized in a mortar with 3 ml 0.1 M potassium phosphate buffer (pH 6.0). After removal of solid compounds by centrifugation at 4°C and 10000 × g for 10 minutes,

400 µL of the supernatant were subjected to gel chromatography with Sephadex G25 medium (GE Healthcare, Chicago, IL, USA) to remove low molecular weight compounds. Peroxidase activity was determined according to the Worthington Manual (1972). Briefly, the enzyme assay contained in a final volume of 1110 µL, 1095 µL buffer 0.1 M potassium phosphate buffer + 0.003 % (v/v) $H_2O_2$ (pH 6.0), 5 µL enzyme preparation, and 10 µL 1 % (w/v) $o$-dianisidin (Sigma-Aldrich-Aldrich, Vienna Austria) in MeOH.

The activity was determined by measuring the extinction at 460 nm on a DU-65 spectrophotometer (Beckman Instruments, Brea, CA, USA) in intervals of 30 s for a period of 6 minutes. The activity was calculated from the slope in the initial linear portion of the reaction progress curved using an extinction coefficient of $1.13 \times 10^4\ M^{-1}\ cm^{-1}$ for oxidized $o$-dianisidine (Worthington manual, 1972). The protein content was determined by a modified Lowry procedure (Sandermann and Strominger, 1972) using bovine serum albumin as a standard. All measurements were performed in two technical replicates.

For the determination of the antioxidant capacity and the TPhen, the material was lyophilised and homogenized by grinding to fine powder in a mortar. 0.25 g of the lyophilised powder was extracted with 3 ml distilled water for 1 hour in a cooled water bath during sonication. After centrifugation for 5 minutes at 4°C and 10000 × g, the supernatant was filtered through a Chromafil AO-20/25 polyamide filter (Roth, Karlsruhe, Germany).

The TPhen was determined as described (Wootton-Beard et al., 2011) with some modifications. Briefly, 100 µL of the aqueous solution was mixed with 6 mL distilled water and 500 µL Folin Ciocalteu Reagent (Sigma-Aldrich, Vienna, Austria) (1:1 v/v with distilled water). After equilibration for 8 minutes, 1.5 ml 20 % $Na_2CO_3$ (w/v) and 1.9 ml distilled water were added and the mixture was incubated at 40°C for 30 minutes. The TPhen was obtained by measuring the absorbance of the mixture at 765 nm using a freshly prepared standard curve obtained with gallic acid. The results were expressed as µg gallic acid equivalents per g sample. All measurements were performed in technical triplicates.

The in vivo antioxidant activity was determined with *Saccharomyces cerevisiae* ZIM 2155 as model system following the procedures described in Slatnar et al. (2012), which estimates intracellular oxidation by fluorometrical measurements using the ROS-sensitive dye 2',7'-dichlorofluorescin ($H_2DCF$). 100 µl of the aqueous samples were incubated with 10 mL yeast suspension at their stationary phase in phosphate buffered saline (PBS, Merck KGaA, Darmstadt, Germany) at a density of $10^8$ cells/suspension at 28°C and 220 rpm for 2 h. After a centrifugation step at room temperature for 5 minutes at $14000 \times g$, the pellet was washed three times with 50 mM potassium phosphate buffer (pH 7.8) and was finally resuspended in 9 volumes of 500 µL 50 mM potassium phosphate buffer (pH 7.8) and incubated for ten minutes at 28°C and 220 rpm in the dark. After addition of 10 µL $H_2DCF$ (1 mM stock solution in 96 % ethanol), the mixture was incubated for further 30 minutes at 28°C and 220 rpm. The fluorescence of the yeast cell suspensions was measured at a GloMax® Multi Microplate Reader (Promega, Walldorf, Germany) using excitation and emission wavelengths of 490 and 520 nm, respectively. Values of fluorescence intensity were measured against a blank, in which the sample was replaced with water. Data are expressed as relative fluorescence intensity, where the values obtained with the blank are defined as 1. Values lower than 1 indicate a higher antioxidant activity than the blank (Slatnar et al., 2012). All measurements were performed in two technical replicates.

**2.4 Statistical analyses**

Emission rates, physiological parameters, means and standard deviation were calculated with Matlab (MATLAB and Statistics Toolbox Release 2017a, The MatWorks, Inc., Natick, MA, United States). All leaf gas exchange ($CO_2$ and $H_2O$) and BVOC flux measurements collected over the seven-day period for the set DS and DS×OS were aggregated into four ranges of SWP (R1: 0.00 to -1.40 MPa; R2: -1.45 to -2.85 MPa; R3: -2.90 to -4.30 MPa, R4: -4.35 to -6.00 MPa) to perform statistical analysis using the Wilcoxon rank sum test. To test for significant differences in the biochemical markers a one-way ANOVA test was used. For both tests p-values below 0.05 were considered significant.

**3 Results and Discussions**

**3.1 Stomatal closure and net photosynthesis**

SWP was measured daily and used as a drought stress indicator to study the evolution of *Q. robur* under continuously increasing drought condition. All six trees began the experiment with a high to moderate mean SWP of -0.9 MPa (Brüggemann and Schnitzler, 2002) and reached low values in the order of -5.5 MPa after seven days of continuously increasing drought stress. Mean and standard deviation of stomatal conductance, net photosynthesis, leaf temperature and SWP as well as notes for statistically significant differences are summarized in Table 1 for the four drought stress ranges defined in 2.4. The mean stomatal conductance ($g_S$) of DS×OS was 20.2 mmol $m^{-2} s^{-1}$ in R1 and decreased to 6.8 mmol $m^{-2} s^{-1}$ in R2 (Table 1). For DS it was 42.4 mmol $m^{-2} s^{-1}$ in R1 and decreased to 6.6 mmol $m^{-2} s^{-1}$ in R2. For both sets the reduction of $g_S$ and SWP between R1 and R4 was significant (p-value 0.02 and 0.05 for DS and DS×OS respectively). R1, shown in Fig. 2 (a), includes values of trees fumigated with ozone (DS×OS) from the first and the second day of analysis, because, for this set, SWP hadn't changed much during these two days. Differently, for DS, R1 includes only measurements of the first day. The values collected during the second day of analysis, for the set DS, is assigned to R2, because we observed a decreased of SWP between the first and second day of measurement. This shows that trees of DS×OS closed their stoma quickly at higher stem water potential after

the first ozone fumigation session, and confirms what was reported in other studies that moderate ozone concentrations can induce partially closed stomata (Khatamian et al., 1973; Farage et al., 1991; Wittig et al., 2007). A partial stomatal closure prevented excessive water loss through stomatal openings (Pinheiro and Chaves, 2011; McDowell et al., 2008; Allen et al., 2010) during drought stress, and enhanced the closure with ozone allowing DS×OS plants to better survive the increased drought. Kobayashi et al. (1993) considers the interactive effects of $O_3$ and drought stress using a growth model of soybean, finding that ozone fumigation reduces or postpones drought stress, similar to the findings of this experiment.

Figure 2 (b) shows a decrease of net photosynthesis ($A$) with the increase of the stress for both set, especially between R1 and R2, whereas the values in R3 and R4 are close to zero. In R1, $A$ presented the same differences exposed for $g_S$ between the sets. Our results are different from the finding of Tjoelker et al. (1995) and Paoletti (2005), where stomatal conductance and photosynthesis are shown to decouple at moderate ozone exposure due to direct damage to biochemical carboxylation, caused by chronic ozone exposure.

The ratio of $C_{BVOCs}$ and $C_A$ is shown in Fig. 3. IS, the dominant BVOC (averagely 96 % of the total emissions), mean standardized IS emissions of DS×OS treated plants were consistently higher in all SWP ranges compared to DS alone (Fig. 4), thus showing the difference between DS and DS×OS in $C_{BVOCs}/C_A$ in the highest SWP ratio range. Initially, at low drought stress (R1), 3-7 % of the assimilated carbon was lost as emitted BVOC, which matches findings in other studies (Sharkey et al., 1991; Baldocchi et al., 1995; Monson and Fall, 1989; Fang et al., 1996), showing that ~2 % of carbon assimilated is lost as IS ($C_{IS}/C_A$) under unstressed conditions and at 30°C. As $CO_2$ assimilation rate decreased quickly, and BVOC emission (especially isoprene emission) stayed elevated the ratio of lost vs. fixed carbon increased to 20 % for DS and 16 % for DS×OS in R2. Pegoraro et al. (2004) reported a carbon loss in the order of 50 % for SWP of -2 MPa, in a drought experiment with *Quercus virginiana*. In R3, the increasing stress corresponded to ratios of 0.7 and 1.03 for DS and DS×OS respectively. Alternative carbon sources for isoprene biosynthesis under drought stress are thus proposed for DS×OS. For example, extra-chloroplastic origin or chloroplastic starch (Karl et al., 2002; Kreuzwieser et al., 2002; Funk et al., 2004; Affek and Yakir, 2003; Schnitzler et al., 2004; Rosenstiel et al., 2003) can sustain carbon sources for isoprene production. At very high drought stress (R4) this ratio decreased again to 0.4 in DS and 0.8 in DS×OS.

### 3.2 BVOCs emissions

To give a general overview on BVOC emissions for both sets Fig. 5 (a) and (b) show the total mass spectra ranging from 40-220 amu for the first and last day of measurement for DS and DS×OS respectively. Figure 5 (c) shows relative changes of the mass spectra between the first and last day of measurements. The mass range 80–110 amu, hosting many mass to charge ratios associated with GLVs, showed the strongest difference between the two sets. Plants exposed to ozone and drought stress (DS×OS) exhibited smaller increases in this mass range compared to drought stressed (DS) plants. Changes in emissions or lack thereof for IS, MT, SQT and stress related BVOCs are investigated in further detail below and are summarized in Table 2.

### 3.2.1 Isoprene emissions

*Q. robur* is generally classified as a high IS emitting (Benjamin and Winer, 1998; Lehning et al., 2002) and medium to low MT and SQT emitting species (Owen et al., 1997; Karl et al., 2009; Steinbrecher et al., 2009). IS emitted by plants and synthesized by the enzyme isoprene synthase (Silver and Fall, 1991) and via the 2-methylery-thritol 4-phosphate (MEP) pathway (Lichtenthaler et al., 1997; Schwender et al., 1997; Lichtenthaler, 1999) in chloroplasts (Wildermuth and Fall, 1996; 1998). Figure 4 shows standardized isoprene emissions ($IS_S$) as a function of drought stress for all investigated trees. In the range of SWP R1 the plants were in a low- to no-water-stress condition (Brüggemann and Schnitzler, 2002). Whereas $g_S$ and $A$ (Fig. 2 (a),(b)) decreased rapidly with increasing drought stress and bottom out at -3 MPa, isoprene emissions decreased

much slower reaching close to zero emissions at R4. $IS_S$ in R1 was 12.8 nmol $m^{-2}$ $s^{-1}$ and 18.0 nmol $m^{-2}$ $s^{-1}$ for DS and DS×OS respectively. In R4 the mean $IS_S$ was 1.7 nmol $m^{-2}$ $s^{-1}$ for DS and 3.9 nmol $m^{-2}$ $s^{-1}$ for DS×OS.

Given that $IS_S$ emissions remain higher in DS×OS for R1 and R2, compared to DS suggests that overall isoprene production within the leaves must have remained high in response to ozone. High IS fluxes due to ozone treatment are also reported in other studies (Fares et al., 2006; Velikova et al., 2005; Kanagendran et al., 2018).

An increase in IS with moderate stress was observed by Pegoraro et al. (2004) and Beckett et al. (2012), who related this finding to an increase in leaf temperatures as a consequence of stomatal closure. In contrast, a no significant increase was observed in the leaf temperatures, suggesting IS emissions of DS×OS in R2 being a result of a temperature-independent isoprene production.

The decrease of $A$ with decreasing SWP, particularly at mild drought stress (Fig. 2 (b)), is much more pronounced than the decrease of $IS_S$ emission rates (Fig. 4). Similar results are found for leaf level measurements of *Q. robur* (Brüggemann and Schnitzler, 2002), *Populus alba* (Brilli et al., 2007) and *Quercus virginia* (Pegoraro et al., 2004) as well as on the ecosystem scale in the Ozark region in the central U.S. (Seco et al., 2015).

Even though the rate of photosynthetic carbon assimilation declined much faster under drought than IS, a substantial decline of IS was also seen as drought progressed. Drought stress has been found to be one of the stronger influencing factors affecting photosynthesis but had often only limited influence on IS emission rates (Tingey et al., 1981; Sharkey and Loreto, 1993; Fang et al., 1996).

In young hybrid poplars (*Populus deltoides* cv. 55/56 x P. *deltoides* cv. Imperial), the combined application of elevated ozone and drought decreases isoprene emission, whereas drought alone increases the emission, and ozone alone decreases it (Yuan et al., 2016).

Studies report that volatile isoprenoids strengthen cellular membranes, thus maintaining the integrity of the thylakoid-embedded photosynthetic apparatus and have a generic antioxidant action by deactivating ROS around and inside leaves and thus indirectly reduce the oxidation of membrane structures and macromolecules (Singsaas et al., 1997; Loreto and Velikova, 2001; Affek and Yakir, 2002; Loreto and Schnitzler, 2010; Velikova et al., 2012).

### 3.2.2 Terpenoid emissions

Monoterpenes (MT) and sesquiterpenes (SQT), other classes of isoprenoids, are synthesized through the condensation of isoprene units (allylic isomer dimethylallyl diphosphate, DMAPP and isoprenyl diphosphate, IPP) (Ruzicka, 1953; Cheng et al. 2007). Geranyl diphosphate (GDP) is the precursor of all MT isomers. GDP is formed from IPP and DMAPP driven enzymatically by GDP synthase (Mahmoud and Croteau, 2002). Farnesyl diphosphate (FDP) synthases adds two molecules of IPP to DMAPP for the formation of the SQT precursors, $C_{15}$ diphosphate (Cheng et al., 2007). Figure 6 shows $MT_S$ (a) and $SQT_S$ (b) as a function of SWP. Mean $MT_S$ for DS and DS×OS were $1.0 \times 10^{-2}$ nmol $m^{-2}$ $s^{-1}$ and $3.6 \times 10^{-2}$ nmol $m^{-2}$ $s^{-1}$ respectively at R1. With the increase of drought stress (R3) DS×OS decreased to $1.5 \times 10^{-2}$ nmol $m^{-2}$ $s^{-1}$ while DS emissions remained stable ($1.0 \times 10^{-2}$ nmol $m^{-2}$ $s^{-1}$). For higher drought stress (R4) both sets showed an increase in MT emissions reaching $3.3 \times 10^{-2}$ nmol $m^{-2}$ $s^{-1}$ for DS and $4.7 \times 10^{-2}$ nmol $m^{-2}$ $s^{-1}$ for DS×OS.

Loreto et al. (2004) demonstrated that ozone can stimulate the emission of monoterpenes in *Q. ilex*, but that ozone has no effect on photosynthesis nor on any other physiological parameter, when Mediterranean oak plants are exposed to mild and repeated, as well as acute ozone stress.

In this experiment MT emissions from *Q. robur*, increased in DS and DS×OS trees. In the case of DS, there was a positive effect of drought, with a significant increase in MT emissions, although there was a drastic decrease of IS emissions when the

water deficit was severe. These observations contrast those by Llusiá and Peñuelas (1998) for *Q. coccifera* reporting a decrease of MT emissions under severe drought conditions. This could be due to the fact that in the case of *Q. coccifera* no specific terpene storage structures are present in leaves, while they are present in *Q. robur* (Karl et al., 2009).

In both sets $SQT_S$ emissions remained close to zero down to a SWP of -3 MPa. $SQT_S$ emissions increase with increasing drought stress reaching a mean value of $1.4 \times 10^{-2}$ nmol m$^{-2}$ s$^{-1}$ for DS and $3.5 \times 10^{-2}$ nmol m$^{-2}$ s$^{-1}$ for DS×OS in R4. The increase of $SQT_S$ in the set with ozone began one day later than in the set without ozone fumigation.

Stress on plants can induce SQT emissions (Toome et al., 2010; Maes and Debergh, 2003; Ibrahim et al., 2006). Ormeño et
al. (2007) observe a reduction of sesquiterpenes with drought stress for a variety of plant species including *Q. coccifera*. For *Q. robur* we see an increase of SQT emissions under conditions of severe drought.

The release of SQT from leaves can be triggered when plants face stress due to oxidative processes in leaves, indicating that damaging effects inside the plants start to occur (Beauchamp et al., 2005; Bourtsoukidis et al., 2012). Unlike MT, SQT don't provide an additional barrier to plant damage during severe water stress (Palmer-Young et al., 2015). This is due to their
different physico-chemical characteristics and the different pathways that produce them (Niinemets et al., 2004; Umlauf et al., 2004). In the case of SQT emissions, the parallel occurrence of two stresses (ozone and increased drought) generally led to an increase in emissions. In fact, the higher SQT emissions in DS×OS compared to DS may have been due to ozone, similar to those reported in Beauchamp et al. (2005).

### 3.2.3 GLV and SHIKIMATE emissions

GLVs are released once the membrane is injured independently of the stress that caused the damage (Heiden et al., 2003). The release of GLVs is related to the degree of damage, and high emissions are linked to high membrane degradation (Fall et al., 1999; Beauchamp et al., 2005; Behnke et al., 2009).

In this experiment, the ΣGLV increased for both sets in R4 (Fig. 7 (a)). Within ΣGLV m/z 99.080, attributable to hexenal isomers, showed the strongest increase in DS (mean value of m/z 99.080 in R4 was 68 % of the Σ GLV emission). Within the
cascade of GLV production, (E)-2-hexenal and (Z)-3-hexenal are typically the ones appearing first (Fall et al., 1999). DS×OS, on the other hand, showed an increase in Shikimate compounds (Fig. 7 (b)) at SWP < -3 MPa, DS showed similar but less pronounced trend. The ΣShikimate was dominated by Methyl Salicylate (MeSa) across the entire SWP range for DS and in R1-R3 for DS×OS. R4 of DS×OS was dominated by m/z 95.050 (matching the exact mass of protonated phenol, $C_6H_7O^+$). MeSa is considered as a volatile stress signaling molecule from plants (Karl et al., 2008). High emissions of MeSa
are also found in the case of the tobacco plant (*Nicotiana tabacum* L. cultivars) in both $O_3$ sensitive and $O_3$ tolerant, exposed to ozone at high concentrations (Heiden et al., 1999; Beauchamp et al., 2005).

Observing the increase in GLV emissions in DS and Shikimate emissions in DS×OS was important to understand how ozone affected the *Q. robur* trees exposed to drought stress. The impact of exposure to high ozone concentrations on ROS production
was not significant and not associated with membrane lesions in Pellegrini et al. (2019). In this experiment, GLV emissions in R4 were no significantly different from R1, with low values in ozone treated plants (DS×OS), while plants that were exposed to drought only (DS) exhibited higher emissions, with a significant increase of GVL emissions between R1 and R4 (Table 2). The observations of this experiment can be interpreted such that plants did not suffer from detrimental effects due to acute ozone exposure yet (e.g. Beauchamp et al., 2005), but that mild ozone exposure can potentially delay effects of drought stress
and help maintain membrane structure and integrity.

The activation of an efficient free radical scavenging system can minimize the adverse effects of a general peroxidation (Miller et al., 1999). This was not the case in DS, where exposure to severe water stress alone led to an increase of GLV emissions

suggesting the onset of physical membrane damage, as the enhancement of the lipoxygenase activity, in accordance with other studies (Ebel et al., 1995; Wenda-Piesik, 2011). In addition to the lipoxygenase and hydroperoxide lyase systems producing GLVs, the phenylpropanoid pathway signals plant responses to stimuli induced by abiotic factors (Dixon and Paiva, 1995; Baier et al., 2005; Heath, 2008; Vogt, 2010), but drought stress alone does not induce the phenylpropanoid pathway in *Q. robur* (Pellegrini et al., 2019).

On the other hand, DS×OS, showed a small increase of GLV only at the highest stress level. We take this to indicate that ozone has the potential to inhibit drought stress damage and therefore the emissions of GLV, by stimulating the phenylpropanoid pathway to form an antioxidant protection for chloroplasts (Pellegrini et al., 2019). The GLV emissions in DS×OS are initially inhibited during of the onset of drought. While ozone fumigation initially inhibits the activation of the lipoxygenase and the hydroperoxide lyase pathway indirectly, these pathways are clearly triggered during the progression of severe drought stress (R4) (Heiden et al., 2003; Matsui, 2006). Cabané et al. (2004) report that, in poplar leaves, ozone exposure not only stimulate the enzymes of the phenylpropanoid pathway, but also the activity of the enzyme SHDH of the shikimate pathway that yield TPhe in fully developed leaves.

To better understand the emissions of GLVs and Shikimate volatiles, we looked at antioxidant capacity, total phenol content and peroxidase activity summarized in Table 3. No significant differences were found for antioxidant capacity between the sets DS, DS×OS and their corresponding references C, and OS. However, it appeared that the OS had the highest oxidizing capacity. TPhen in the fully developed leaves was significantly higher in the two groups experiencing drought stress (DS, DS×OS) than in those with no drought stress (C, OS). Pellegrini et al. (2019) found, a significant difference in TPhen content in well-watered plants with the increase of ozone, and a decrease at moderate drought and no significant influence of ozone on TPhen during severe drought in *Q. robur*. The results of our study showed no significant decrease in TPhen due to ozone fumigation both in well-watered and severe drought condition (R4) (OS, DS×OS). Peroxidase activity analysis did not show significant differences between the four sets. This is in accordance with the finding of Schwanz and Polle (2001) who found that unspecific peroxidase activities are not affected by drought stress in *Q. robur*.

## 4. Conclusions

The changes in BVOC emissions of *Q. robur* subject to continuously increasing drought were investigated and differences in the drought progression were observed in plants with and without ozone fumigation. Stomatal conductance and net photosynthesis showed a fast reaction to increasing drought closing stomata and reducing $CO_2$ uptake strongly. $IS_S$ emissions, on the other hand, stayed high down to a SWP of -3 MPa and then decreased gradually. We consider that leaves must have maintained a high production of IS to sustain similar emissions compared to a SWP of -2 MPa. $MT_S$ and $SQT_S$ emissions increased under high drought stress. Plants that were subject to one hour of ozone fumigation (~100 ppbv) every day in addition to reduced watering showed lower stomatal conductance at mild drought stress compared to those with no ozone fumigation, and consecutively the effect of drought was slowed down. The Shikimate pathway, producing antioxidants, was stimulated earlier in the set with ozone. The combination of (i) sustained isoprene emissions, (ii) increase of antioxidants due to the higher stimulation of the two pathways (phenylpropanoid and shikimate) and (iii) early closure of the stomata resulted in a longer endurance of drought stress in the set exposed to ozone. Therefore, we conclude that fumigation with moderately high ozone levels (~100 ppbv) decelerated the effect of drought in *Q. robur*. Overall *Q. robur* leaves appeared very resistant to drought stress. Consequently GLVs indicating cell damage were only emitted at SWP < -5 MPa.

As seasonal drought events and elevated ozone concentrations often occur in parallel in mid latitudes (Löw et al., 2006; Panek et al., 2002) it is important to study their combined stress effects. In this study we observe that a combination of stresses can lead to opposing feedbacks that alter BVOC emissions. These effects are compound specific and reflect biochemical changes in the plant.

## Author contribution

AP, LK, TGK, GW, HH drafted the manuscript, which was edited by all co-authors. Laboratory work was performed by AP, LK, ACF, MG, TGK, HS and JG. AP, LK, ACF and HH analyzed and interpreted the data.

## Competing interests

The authors declare that they have no conflict of interest.

## Acknowledgments

This work was supported by the Vienna Science and Technology Fund (WWTF, project number: ESR17-027). In addition AP was supported by a doctoral grant fellowship of the LFU. We are grateful to Polona Jamnik for kindly providing *Saccharomyces cerevisiae* ZIM 2155 from the Culture Collection of Industrial Microrganisms (ZIM) of the Biotechnical Faculty of University of Lubljana, Lubljana, Slovenia. Support in the analysis of the leaves by Silvija Marinovic and Michael Kurta at TU Wien is also gratefully acknowledged. We would also like to thank Astrid Mach-Aigner (Research Group:

Synthetic Biology and Molecular Biotechnology at the Institute of Chemical, Environmental and Biocscience Engineering,TU Wien) for kindly offering access to the GloMax®-Multi Microplate Reader.

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

**Tables**

**Table 1: Mean and standard deviation in parenthesis for stomatal conductance (gs), net photosynthesis (A), leaf temperature (Tleaf) and stem water potential (SWP) divided into four ranges of SWP (R1: 0.00 to -1.40 MPa; R2: -1.45 to -2.85 MPa; R3: -2.90 to -4.30 MPa, R4: -4.35 to -6.00 MPa).**

| | R1 | | R2 | | R3 | | R4 | |
|---|---|---|---|---|---|---|---|---|
| | DS | DS×OS | DS | DS×OS | DS | DS×OS | DS | DS×OS |
| $g_s$ [mmol m$^{-2}$ s$^{-1}$] | **42.4 (28.9)[a]** | 20.2 (13.8)[c] | 6.6 (4.9) | 6.8 (2.7) | 3.8 (0.8) | 3.3 (0.4) | **3.9 (0.7)[a]** | 2.9 (0.1)[c] |
| $A$ [μmol m$^{-2}$ s$^{-1}$] | **3.38 (2.08)[a]** | 1.99 (1.37)[c] | 0.58 (0.78) | 0.52 (0.36) | 0.08 (0.07) | 0.05 (0.02) | **0.10 (0.10)[a]** | 0.02 (0.004)[c] |
| $T_{leaf}$ [K] | 302.3 (1.9) | 303.0 (1.7) | **302.5 (1.0)[b]** | **301.3 (0.6)[b]** | 302.1 (0.9) | 302.6 (1.2) | 301.1 (1.1) | 302.7 (0.2) |
| SWP [MPa] | **-0.9 (0.2)[a]** | -0.9 (0.1)[c] | -2.0 (0.1) | -2.3 (0.2) | -3.3 (0.2) | -3.6 (0.5) | **-5.5 (0.5)[a]** | -5.4 (0.7)[c] |

**Values in bold marked with (a) indicate a significant (p-values<0.05) differences between R1 and R4, (b) indicate a significant difference between the set under drought stress (DS) and the set under drought stress with ozone treatment (DS×OS). Values marked with (c) indicate close to significant differences with p- values of 0.05-0.06 between R1 and R4.**


**Table 2: Mean and standard deviation for standardize isoprene emissions (IS$_S$), standardized monoterpenes emissions (MT$_S$), standardized sesquiterpenes emissions (SQT$_S$), sum of GLV (ΣGLV) and sum of Shikimate (ΣSHIKIMATE) for each set divided by range of stem water potential (SWP) (R1: 0.00 to -1.40 MPa; R2: -1.45 to -2.85 MPa; R3: -2.90 to -4.30 Mpa, R4: -4.35 to -6.00 MPa).**

| | R1 | | R2 | | R3 | | R4 | |
|---|---|---|---|---|---|---|---|---|
| | DS | DS×OS | DS | DS×OS | DS | DS×OS | DS | DS×OS |
| IS$_S$ [nmol m$^{-2}$ s$^{-1}$] | **12.8 (2.0)[a]** | 18.0 (7.3)[c] | **8.6 (3.8)[b]** | **17.3 (4.1)[b]** | 6.9 (2.4) | 10.6 (4.4) | **1.7 (0.9)[a]** | 3.9 (2.6)[c] |
| MT$_S$ [nmol m$^{-2}$ s$^{-1}$] | **0.010 (0.002)[a]** | 0.036 (0.026) | 0.009 (0.004) | 0.023 (0.013) | 0.010 (0.002) | 0.015 (0.010) | **0.033 (0.014)[a]** | 0.047 (0.012) |
| SQT$_S$ [nmol m$^{-2}$ s$^{-1}$] | **0.002 (0.001)[a]** | 0.002 (0.001)[c] | 0.003 (0.002) | 0.001 (0.001) | 0.005 (0.003) | 0.007 (0.008) | **0.014 (0.005)[a]** | 0.035 (0.007)[c] |
| ΣGLV [nmol m$^{-2}$ s$^{-1}$] | **0.002 (0.001)[a]** | 0.003 (0.004) | **0.004 (0.005)[b]** | **0.001 (0.003)[b]** | 0.002 (0.002) | 0.001 (0.001) | **0.032 (0.045)[a]** | 0.009 (0.010) |
| ΣSHIKIMATE [nmol m$^{-2}$ s$^{-1}$] | 0.001 (0.001) | 0.003 (0.001)[c] | 0.001 (0.002) | 0.003 (0.003) | 0.004 (0.002) | 0.008 (0.012) | 0.003 (0.001) | 0.009 (0.001)[c] |

**Values in bold marked with (ª) indicate a significant (p-values<0.05) differences between R1 and R4, (ᵈ) between R2 and R3, (ᵇ) indicate a significant difference between the set under drought stress (DS) and the set under drought stress with ozone treatment (DS×OS). Values marked with (ᶜ) indicate close to significant differences with p-values of 0.05-0.06 between R1 and R4.**


**Table 3: Mean and standard deviation of antioxidant capacity, total phenol content (TPhen), peroxidase activity for well-watered sets with (OS) and without (C) ozone treatment, and for sets under severe drought stress with (DS×OS) and without (DS) ozone treatment after seven days of measurements.**

| | Antioxidant Capacity [(F ODSample$^{-1}$) (F ODControl $^{-1}$) $^{-1}$] | TPhen [grGAEequiv. kg$^{-1}$ (DW)] | Peroxidase Activity [μmol s$^{-1}$ kg$^{-1}$(DW)] |
|---|---|---|---|
| **C** | 0.9 (0.1) | **35.6 ( 11.7)[e]** | 0.9 (0.7) |
| **OS** | 0.8 (0.04) | **25.8 (11.7) [f]** | 0.6 (0.3) |
| **DS** | 1.0 (0.1) | **86.5 (24.1)[e]** | 0.9 (0.3) |
| **DS×OS** | 1.0 (0.1) | **77.1 (9.2) [f]** | 0.8 (0.4) |

**Values in bold marked with (ᵉ) represent values with significant (p-value <0.05) differences between C and DS, (ᶠ) represent values with significant differences between OS and DS×OS.**

**Figures**

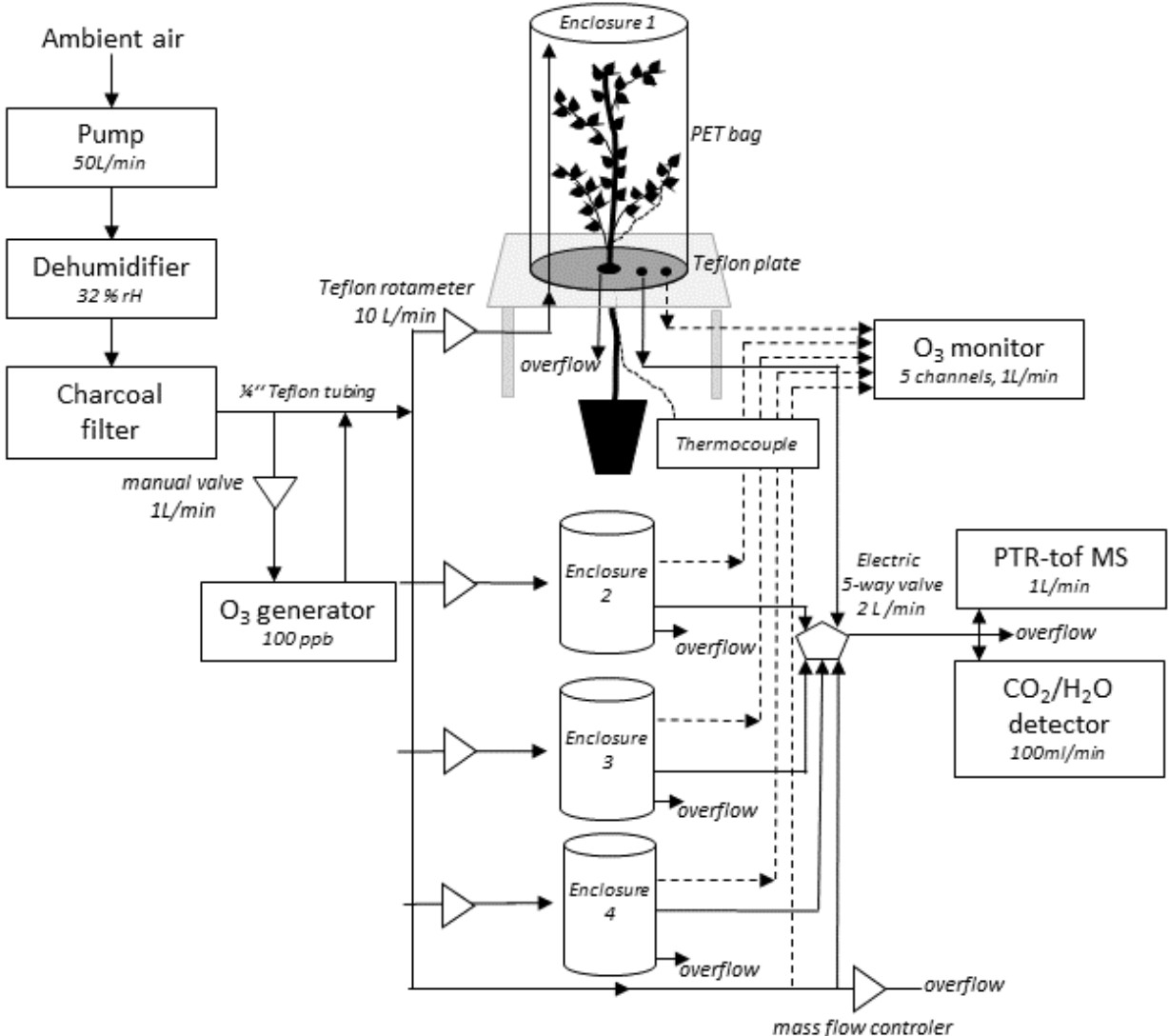

Figure 1: Scheme of a custom-made plant enclosure and set up of the experiment. In brief, the chambers consisted of a PTFE-covered
bottom plate with an opening mechanisms to insert and seal the plant stem using PTFE-plugs; furthermore, the bottom plate
featured three in- and outlets for gas sampling and ozone exposure; the inlet was raised above the bottom plate to allow for air
mixing. The upper part of the chamber consisted of a transparent, 12-liter PET-bag, holding most of the tree crown. The bags were
tightly sealed towards the bottom plate.

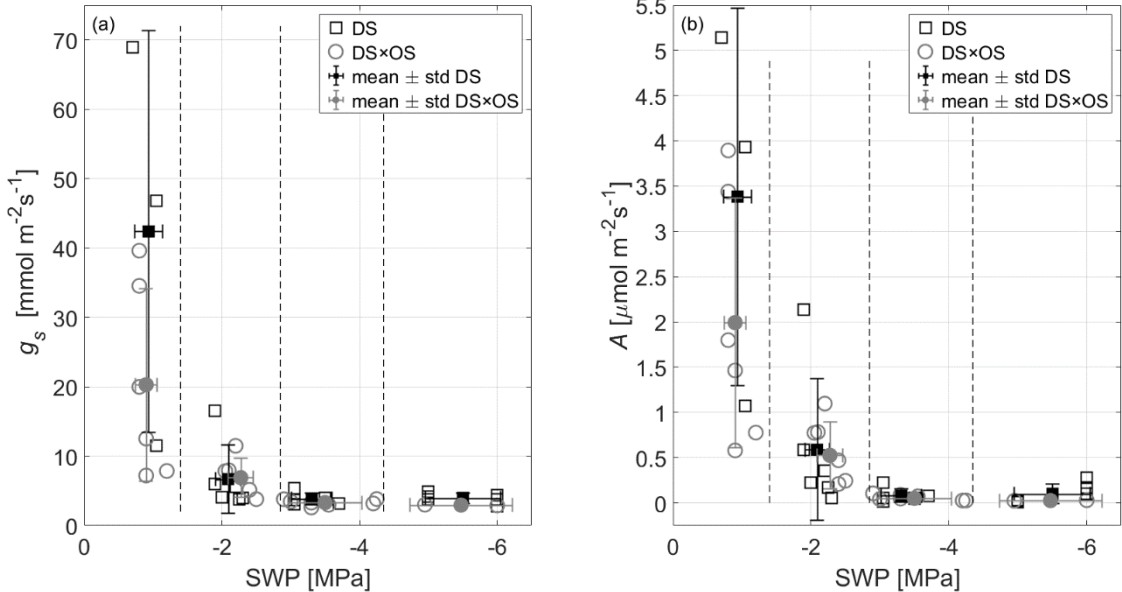


**Figure 2: (a) Stomatal conductance ($g_s$) and (b) net photosynthesis ($A$) of all trees as a function of stem water potential (SWP).** Empty markers represent individual trees where the black squares represent trees out of the set under drought stress (DS) and the gray circles out of the set under drought stress with ozone treatment (DS×OS). Filled squares and circles represent means calculated for each SWP range with the corresponding standard deviation. SWP ranges are separated by vertical dashed lines.

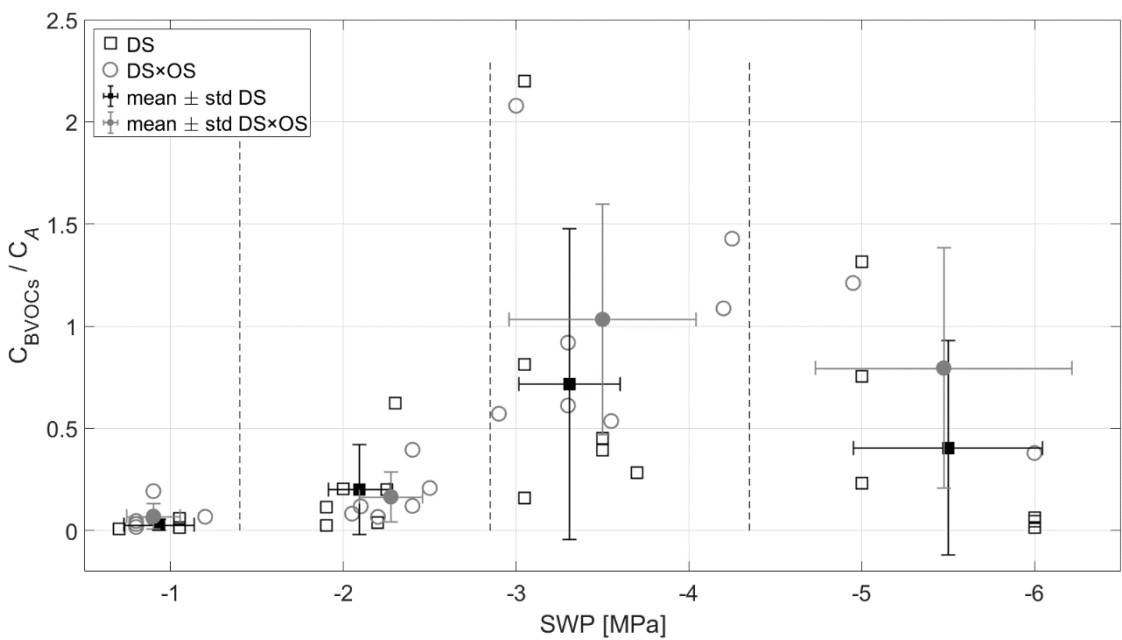


**Figure 3: Ratio of sum of carbon emitted by all analyzed BVOCs ($C_{BVOCs}$) and the sum of carbon uptake via net photosynthesis ($C_A$) versus the stem water potential (SWP).** Empty markers represent individual trees where the black squares represent trees out of the set under drought stress (DS) and the gray circles out of the set under drought stress with ozone treatment (DS×OS). Filled squares and circles represent means calculated for each SWP range with the corresponding standard deviation. SWP ranges are separated

by vertical dashed lines.

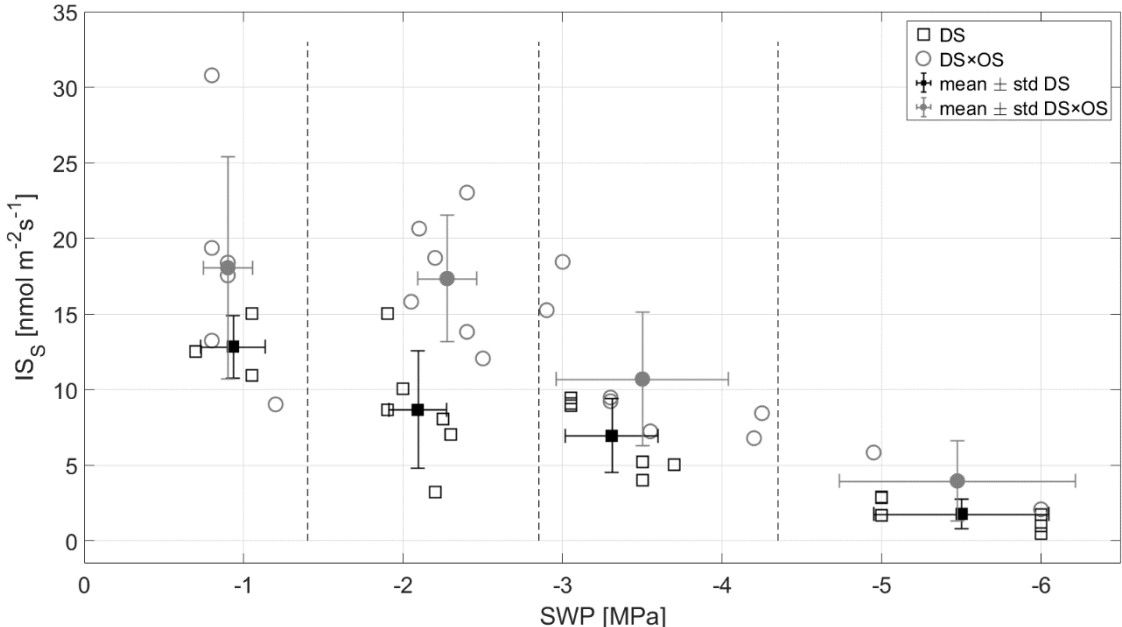

**Figure 4: Standardized isoprene emission (IS$_S$) versus stem water potential (SWP). Empty markers represent individual trees where the black squares represent trees out of the set under drought stress (DS) and the gray circles out of the set under drought stress with ozone treatment (DS×OS). Filled squares and circles represent means calculated for each SWP range with the corresponding standard deviation. SWP ranges are separated by vertical dashed lines.**


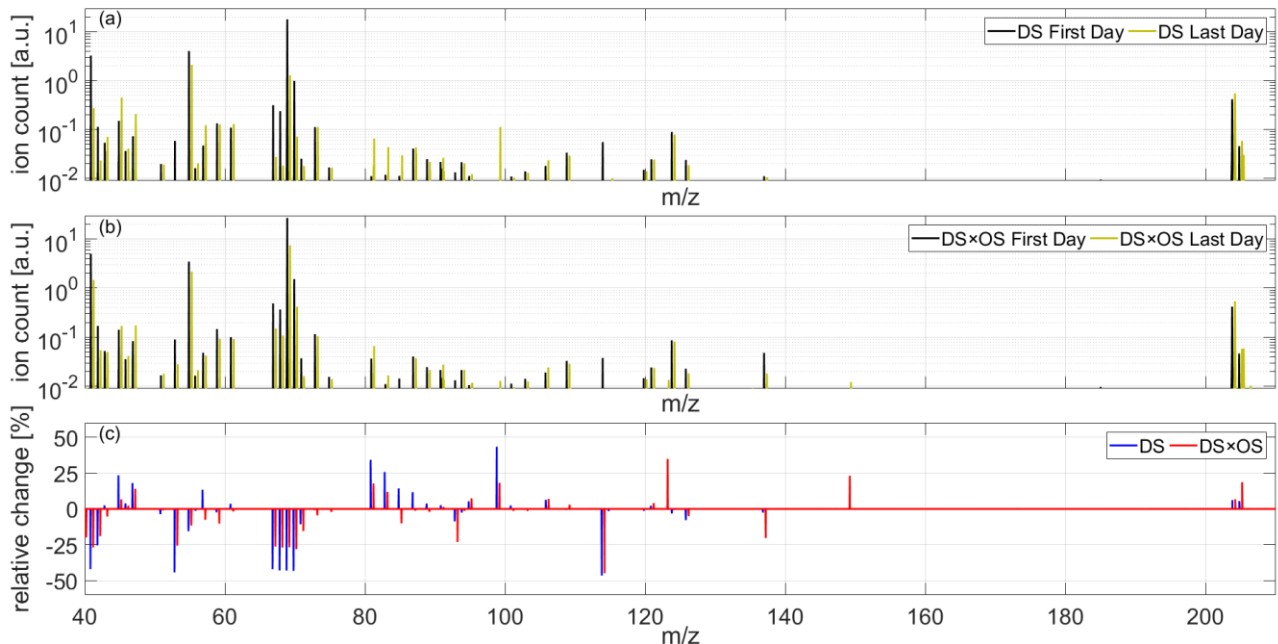

**Figure 5: Mean mass spectra of the set under drought stress (DS) (a) and the set under drought stress with ozone fumigation (DS×OS) (b), on the first (black) and last (yellow) day of measurement. (c) Relative change in the mass spectra between the last and the first day of analysis for DS (blue) and DS×OS (red).**

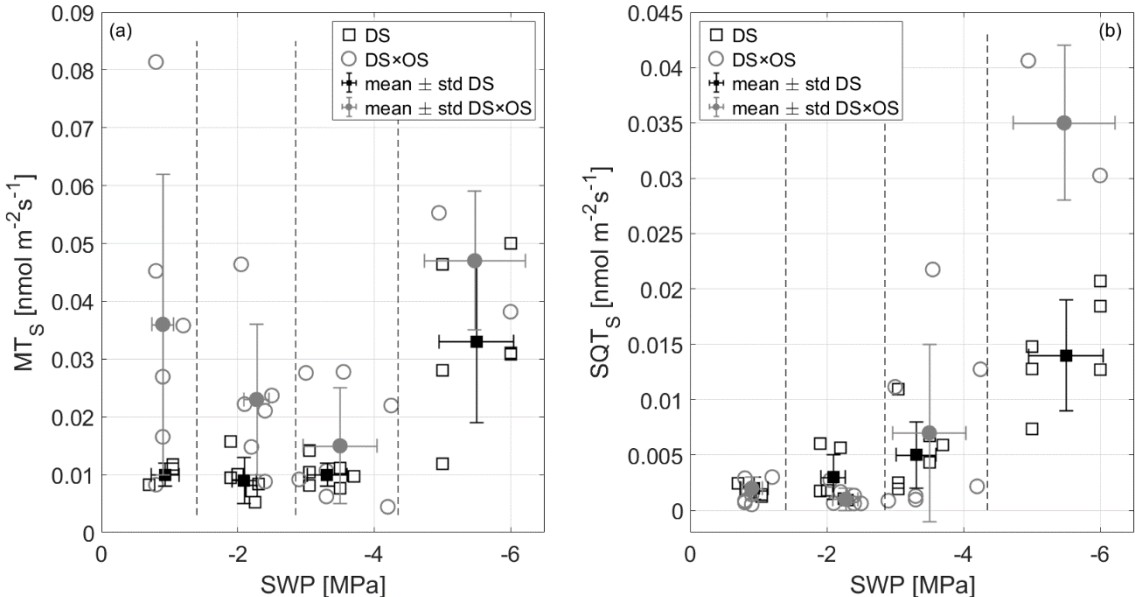

**Figure 6: Standardized monoterpenes (MTS) (a) and sesquiterpenes (SQTS) (b) emissions versus stem water potential (SWP). Empty markers represent individual trees where the black squares represent trees out of the set under drought stress (DS) and the gray circles out of the set under drought stress with ozone treatment (DS×OS). Filled squares and circles represent means calculated for each SWP range with the corresponding standard deviation. SWP ranges are separated by vertical dashed lines.**

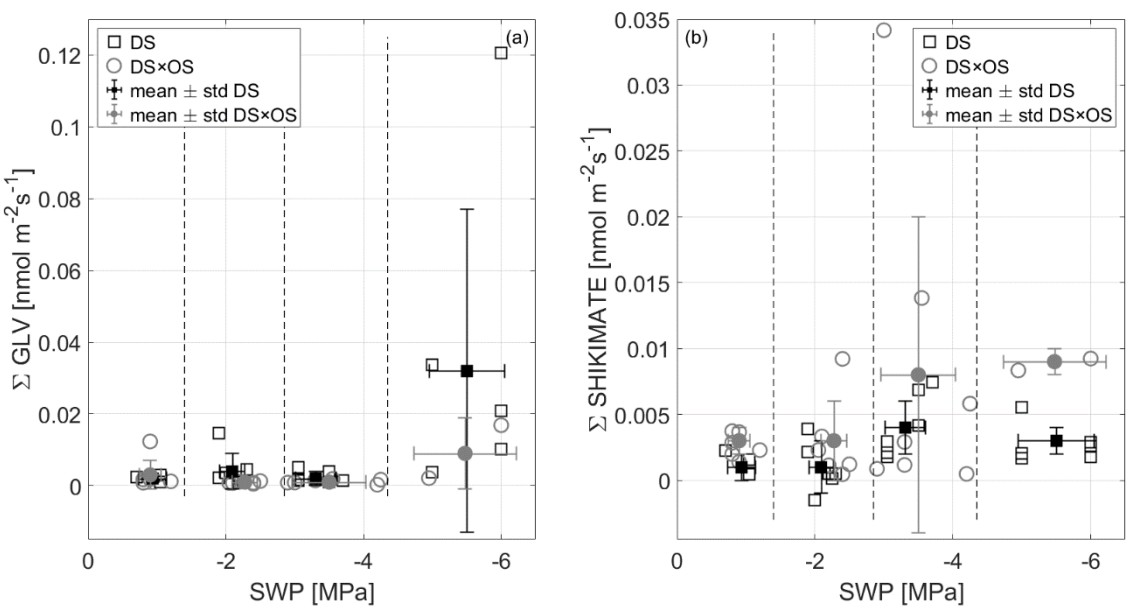

**Figure 7: The sum of green leaf volatiles (ΣGLV) (a) and the sum of Shikimate (ΣShikimate) compound (b) emissions versus stem water potential (SWP). Empty markers represent individual trees where the black squares represent trees out of the set under drought stress (DS) and the gray circles out of the set under drought stress with ozone treatment (DS×OS). Filled squares and circles represent the mean values calculated for each SWP range with the corresponding standard deviation. SWP ranges are separated by vertical dashed lines.**

## Appendix A

**Table A1: Acronyms and experimental conditions used in this experiment.**

| ACRONYMS | |
|---|---|
| *A* | Net photosynthesis ($CO_2$ assimilation rate) |
| **BVOCs** | Biogenetic Volatile Organic Compounds |
| **C** | Control samples without ozone treatment |

| | |
|---|---|
| **DS** | Set under drought stress without ozone treatment |
| **DS×OS** | Set under drought stress with ozone treatment |
| *gs* | Stomatal Conductance |
| **GLVs** | Green Leaf Volatiles |
| **IS** | Isoprene |
| **ISs** | Standardized emissions of Isoprene |
| **MeSa** | Methyl Salicylate |
| **MT** | Sum of Monoterpenes |
| **MT$_S$** | Standardized emissions of MT |
| **O$_3$** | Ozone |
| **OS** | Well-watered control samples with ozone treatment |
| **PTR-ToF-MS** | Proton Transfer Reaction Time of Flight Mass Spectrometer |
| *Q. robur* | *Quercus robur* L. |
| **ROS** | Reactive Oxygen Species |
| **SQT** | Sum of Sesquiterpenes |
| **SQT$_S$** | Standardized emissions of SQT |
| **std** | Standard deviation |
| **SWP** | Stem Water Potential |
| **TPhe** | Total Phenol content |

**EXPERIMENTAL CONDITIONS**

| | |
|---|---|
| **Enclosure Pressure** | 2.386 kPa |
| **Mean leaf temperature** | 29.06°C |
| **Mean PAR** | 1374 µmol m$^{-2}$ s$^{-1}$ |
| **Ozone concentration** | 100 ppb |
| **Standardized temperature** | 30°C |
| **Standardized PAR** | 1000 µmol m$^{-2}$ s$^{-1}$ |

**Table A2: m/z ratio and chemical formula and name of compounds presents in the standard gas mixture used for the daily calibration of the PTR-Tof-MS.**

| m/z ratio | Chemical formula | Compound |
|---|---|---|
| 32.0262 | $CH_3OH$ | Methanol |
| 41.0265 | $C_2H_3N$ | Acetonitrile |
| 44.0261 | $C_2H_4O$ | Acetaldehyde |
| 58.0418 | $C_3H_6O$ | Acetone |
| 72.0574 | $C_4H_8O$ | Methyl Ethyl Ketone (MEK) |
| 78.0469 | $C_6H_6$ | Benzene |
| 92.0625 | $C_7H_8$ | Toluene |
| 106.0782 | $C_8H_{10}$ | Xylenes |
| 120.0939 | $C_9H_{12}$ | 1,2,4-Trimethylbenzene (TMB) |
| 136.1252 | $C_{10}H_{16}$ | a-Pinene |
| 62.0189 | $C_2H_6S$ | Dimethyl sulphide (DMS) |
| 86.0731 | $C_5H_{10}O$ | 2-methyl-3-buten-2-ol (MBO) |
| 134.1095 | $C_{10}H_{14}$ | 1,2,4,5-Tetramethylbenzene |

**Table A3: Measured m/z ratio, chemical formula and tentative assignment of compounds used for the calculation of the sum of BVOCs in $C_{BVOCs}$ / $C_A$.**

| m/z ratio | Chemical formula | Compound |
|---|---|---|
| 33.033 | $(CH_4O)H^+$ | Methanol |
| 45.033 | $(C_2H_4O)H^+$ | Acetaldehyde |
| 47.049 | $(C_2H_6O)H^+$ | Ethanol |
| 57.033 | $(C3H_4O)H^+$ | E-2-Hexenal fragment |
| 57.069 | $(C_4H_8)H^+$ | Butyl |
| 59.049 | $(C_3H_6O)H^+$ | Acetone |
| 61.028 | $(C_2H_4O_2)H^+$ | Acetic Acid |
| 71.049 | $(C_4H_6O)H^+$ | Methyl Vinyl Ketone (MVK) /Methacrolein (MAC) |
| 73.064 | $(C_4H_8O)H^+$ | Methyl Ethyl Ketone (MEK) |
| 79.054 | $(C_6H_6)H^+$ | Benzene |
| 83.085 | $(C_6H_{10})H^+$ | Hexanals/Hexenol fragment |
| 85.101 | $(C_6H1_2)H^+$ | Hexene |
| 87.080 | $(C_5H_{10}O)H^+$ | 2-methyl-3-buten-2-ol (MBO) |
| 93.069 | $(C_7H_8)H^+$ | Toluene/MT fragment |
| 95.050 | $(C_6H_5OH)H^+$ | Phenol |
| 99.080 | $(C_6H_{10}O)H^+$ | Hexenals |
| 101.096 | $(C_6H1_2O)H^+$ | Hexanal |
| 107.049 | $(C_7H_6O)H^+$ | Benzaldehyde |
| 107.073 | $(C_8H_{10})H^+$ | Xylenes |
| 143.107 | $(C_8H_{14}O_2)H^+$ | Hexenylacetate |
| 145.122 | $(C_8H_{16}O_2)H^+$ | Hexylacetate |
| 153.055 | $(C_8H_8O_3)H^+$ | Methyl Salicylate (MeSa) |
| 165.092 | $(C_{10}H_{12}O_2)H^+$ | Eugenol |
| 211.133 | $(C_{12}H_{18}O_3)H^+$ | Jasmonic Acid |
| 225.149 | $(C_{12}H_{20}O_3)H^+$ | Methyl Jasmonate |
| 265.144 | $(C_{15}H_{20}O_4)H^+$ | Abscisic Acid (ABA) |
| 69.070 | $(C_5H_8)H^+$ | Isoprene (IS) |
| 137.133 | $(C_{10}H_{16})H^+$ | Monoterpenes (MT) |
| 205.195 | $(C_{15}H_{24})H^+$ | Sesquiterpenes (SQT) |

925 **Table A4: Mean dry weight and mean specific leaf area for the 20 % of the total analysed leaves of sets DS and DS×OS.**

| | Dry weight [g] | Specific leaf area [m$^2$] |
|---|---|---|
| DS | 1.16 | 0.015 |
| DS×OS | 0.82 | 0.011 |