# Peer review of "Combined effects of ozone and drought stress on the emission of biogenic volatile organic compounds from Quercus robur L."

_Biogeosciences, 2020_

## Referee Comment (RC1) · Anonymous Referee #1 · 24 Aug 2020

This study focuses on the effect of simultaneous drought and ozone stress on emission of biogenic volatile organic compounds from Quercus robur L. The manuscript provides important information about the combined effect of drought and ozone exposure stressors. The finding that these stress factors are not always additive by itself is important, highlighting the need for additional study on the combined effects of multiple stressors. Yet, this notion is not novel (e.g., see references below). Some insights about the enzymatic pathways which were involved in the combined stress effects and the detailed information about the response of different BVOC types to different drought levels and combined stress effects can be useful for future study.

Overall the manuscript is well written. In some cases, important methodological information is missing. I think that the Introduction lacks important explanation about the motivation of addressing your hypothesis on the basis of the key roles that BVOCs play in the troposphere. Specific comments are listed below.

Specific comments Line 20: please specify to what levels.

Line 26: "," should be placed after "Plants" and not after "in".

Line 48 "it has be shown" – please fix.

Line 49: Remove ",".

Line 51: "and longer growing longer seasons" – please rephrase.

Lines 51-54: However, drought can also lead to the opposite effect, as you mention below, which seems to be in conflict with this statement.

Line 87 - The hypothesis of this study is based on the research question whether drought and ozone stressors are additives. It would be good to expand more about 1) previous study of these two stressors in a broader scope (e.g., (Pollastrini et al., 2014;Wittig et al., 2007)) and 2) the importance of currently addressing this specific research question (e.g., how it can help scientists to cope with current assessment/research questions). 3) the state of knowledge with respect to BVOC emission (e.g., (Holopainen and Gershenzon, 2010)).

Line 87 - "different abiotic stresses" – be specific about whether you refer here only to ozone and drought stressors and/or to additional stressors.

Introduction- it will be good to provide information about the role of BVOCs in the troposphere.

Lines 93-94 – " its fast-regulated transpiration rates "- This is not clear. I'm not sure that these two factors are sufficient to result in high tolerance to drought.

Line 106- "air gas exchange" – Do you mean CO2 and H2O exchange- please be specific in this definition throughout the text (e.g., line 119 and elsewhere). Can you elaborate on how did you measured this gas exchange?

I recommend adding the schematic of the experimental design (Fig. A1) to the Materials and Methods. It will help in following your experiments.

Lines 120-121 – please specify if this part still take place in the fitotron.

Line 131- Please define "rH".

Line 132- "assured" – can you explain in detail why it assures so?

Lines 135-136- Irradiation took place also in nighttime? Please specify.

Can you provide details about Background (zero) calibration? e.g., if and what frequency and under what conditions such calibration was performed using the PTR-TOF6000X2?

Line 157 – it is not clear to me what do you mean by "combined calibration uncertainties" and by "a compound specific average experiment sensitivity".

Line 158 – "40-800 pptv" – it is not clear to me why providing this information is important. It may be more useful to specify specific limits of detection to individual compounds (possibly in Table A3).

Lines 161-168 – I suggest to include each individual compound acronym together with its specific m/z (e.g., in parenthesis) and the specific reference, either in the text or in a table.

Line 179 – Why don't you include each of the abbreviations in parenthesis?

Line 189 – Can you elaborate on how the stomatal resistance was measured? Was it measured for each specific leaf or using another approach?

Lines 189-190 – Can you explain why assuming that the boundary resistance is zero

is justified?

Line 197 – " (see below)" – it would be better to specify the specific section number.

Lines 203-204 – "peroxidase and antioxidant capacity, and phenol content" – it is the first time you mention these properties. It would be good to expand on them and why they were measured.

Line 216 – What do you mean by "linear range of..." ?

Line 256 – no need for multiple definition in the main text.

Line 260 – " significant." – can you add a P-value?

Line 274 - no need for multiple definition in the main text.

Line 285 – " (averagely 96 % of the total emissions" – it would be better to provide this information earlier, so the reader will have this in mind when reading the second paragraph in this section.

Line 287 – "carbon loss ratio" - Be more accurate in definition. Do you mean->CIS/CA?

Line 288 –" high drought stress" - Can you specify this in terms of "R"?

Line 304 – Please add "." at the end of the sentence.

Lines 340-341 – "a decrease of MT emissions" – under what conditions?

Line 366- Is this a new paragraph (if so, make it clear and consistent with the rest of the manuscript)?

Lines 385-386 – " by stimulating the phenylpropanoid pathway" – what about the lipoxy-genase and hydroperoxide systems?

Line 396 – " well-watered and severe drought condition" – can you specify these also in terms of R?

Tables 1 and 2 – it is recommended to include the comments below the table. Figure

[Figure]

A1 – Can you include the thermocouple in the figure? What about

Table A3 – add the compound acronyms/names.

References

Holopainen, J. K., and Gershenzon, J.: Multiple stress factors and the emission of plant VOCs, Trends Plant Sci, 15, 176-184, 10.1016/j.tplants.2010.01.006, 2010. Pollastrini, M., Desotgiu, R., Camin, F., Ziller, L., Gerosa, G., Marzuoli, R., and Bussotti, F.: Severe drought events increase the sensitivity to ozone on poplar clones, Environ Exp Bot, 100, 94-104, 10.1016/j.envexpbot.2013.12.016, 2014. Wittig, V. E., Ainsworth, E. A., and Long, S. P.: To what extent do current and projected increases in surface ozone affect photosynthesis and stomatal conductance of trees? A meta-analytic review of the last 3 decades of experiments, Plant Cell Environ, 30, 1150-1162, 10.1111/j.1365-3040.2007.01717.x, 2007.

---

## Referee Comment (RC2) · Anonymous Referee #2 · 7 Sep 2020

The manuscript entitled "Combined effects of ozone and drought stress on the emission of biogenic volatile organic compounds from Quercus robur L." present interesting data about the ecophysiological and volatile emission response of Quercus robur, studying combined stress effects on plants which is a subject that needs to be investigated further. It is well written and fits in the scope of Biogeosciences. However before endorsing publication some revisions must be made. The introduction could be more concise and to the point of the hypothesis. I believe there is too much information at first about terpenoid biosynthesis, which if needed could be explained better in detail in the discussions relating it to the results. When you start talking about temperature as a dominant stress, one asks why you have not used temperature as a treatment, thus I

recommend removing it (just mention it overall and go directly to the drought and ozone issue). The paragraph of the combination of stress is great, but then the biosynthetic information seems misplaced, perhaps do the same as for terpenoid biosynthesis. The objectives must be better explained and put into context in the introduction. Particularly having a paragraph above saying this was already done. Why, having Vitale et al., 2008 and Yuan et al., 2016, we need this study. Additionally, a bit of background about Quercus robur already in the introduction would be interesting, to support why you chose this species (more than a widely distributed isoprene emitting oak species, i.e. is this species going to suffer drought and ozone stress in particular? Why?). Methodology must be better explained. In particular a diagram choosing the number of replicates chosen for each treatment. You say you have 14 trees in total, how are they separated. For instance R4 only has two replicates for DSxOS, why?. Additionally a time series of watering and lack of watering could be expressed in this diagram as well. It is confusing what you use for emission measurements, for biochemical assays and for references. To sum up the methodology must be revised and better explained. Think that the reader must be able to reproduce your methodology. More detailed to be found below. Results and Discussion: Sometimes it looks a bit junky, as here is my results and this is its related reference. I suggest improving the whole section to make clear what have you found differently than literature or else separating results and discussion to aid this task. If you say your results relate to this and that, please expand on how they relate (using your numbers). Additionally use this chance to explain better about the implication and potential of your study (i.e. how the future modelling could change with these new results)

Line 96: where do the 2-year-old trees are coming from? Line 97: What do you mean by soil used by the city gardeners? What brand? Line 97: What brand is the quartz sand from? Line 98: how much fertilizer you put? Line 99: Tulln is a place not a brand... put the brand of the greenhouse or say how did you reach the levels mentioned. Line 100: what do you mean by close by experiment Line 101: Please state better the time of measurements. Line 101: the biochemical assays should also be

explained in the diagram. Line 104: where do you perform the drought stress, in what conditions are the plants? Line 105: I do not understand what do you mean by "maintained by keeping the soil water content at 4-5 vol%". Wasn't this a continuously increasing drought? Was this maintained at all SWP ranges? Then the control plants were at field capacity? Please explain better Line 106: I am really concern with plant acclimation here. As far as I understand the plants are moved ONLY 24 hours before measurements to the climate chamber. Is this enough? Please argue how is this enough. Additionally do the plants stay there or they go back to the greenhouse? I hope they stay in the climate chamber, otherwise it wouldn't be right. Please state. Line 108: what do you mean by mid canopy height? What was the PAR level at the climate chamber? What do you mean by to adapt to constant air temp? Line 111: So DSxOS individuals are fumigated with ozone inside the enclosure while measurements were taken place or prior measurements? Please state. Line 113: why humidity was decrease and temperature increased to maintain the drought strees? Wasn't this maintained by the SWC? Line 115: what's is C and what is OS? Line 119: what do you mean by tree leaf gas exchange? Please state what do you mean by gas exchange and also why not this is tree emission as the branch is also inside the cuvette. Line 121: as far as I understand you maintain the tree during the seven days inside the cuvette? Do you have as many cuvettes or only 4? Please explain better Line 131: why only 370 ppm of $CO_2$, is this realistic to nowadays? Line 145: how did you calibrate for $CO_2$ and $H_2O$? Line 155: please state the compounds inside. Line 155: why did you have to perform calibrations so often? Line 152: do you use an average calibration factor for all compounds? Which is certainly not correct but at least what I interpret from the text. Please state how do you specifically calibrate for GLV, MESA and Sqt. Do you have them in your calibration bottle? Line 161-166: please talk about possible contributors to this mass… how are you sure you can only attribute those signals to the mentioned compounds? Line 193: please name and comment on the calibration of these compounds. Line 254: why high to moderate, any references? 260-263: please rephrase, I just don't get it. 291: I wonder about the Line 303: wouldn't it be better to

say R4 instead of SWP -6MPa? Line 310: please can you mention on how they did not change? Line 334: actually for MT DS there was no significant increase…… Line 373-374: please state the values of low and high Line 404: please do not use the word believe!

―――――――――――――――――

---

## Short Comment (SC1) · 17 Sep 2020

This paper offers a valuable contribution to evaluating the effects on the BVOC emissions from Q. robur of two of the main sources of environmental stress (ozone and drought) which are expected to become more frequent in mid-latitudes in the coming decades. As the authors describe in the Introduction, isoprene emissions have been reported to increase under moderate drought and then decrease as the drought becomes severe. This behaviour is typically ascribed to a decrease in stomatal conductance at the onset of drought, which in turn increases leaf temperature driving higher isoprene emissions. As the the drought becomes severe, a shortage of carbon substrate leads

to a reduction in isoprene synthesis and emissions.

Keeping this conceptual mechanism in mind, it is interesting how the isoprene emissions reported in Fig 3 (drought stress data only) seem only to decrease steadily, without peaking under moderate drought as reported in the literature*. From the leaf temperature values reported in Table 1 it seems clear that leaf temperature in the DS experiments remained relatively constant (within the indicated standard deviations), whereas stomatal conductance decreased rapidly as SWP decreased. Could the authors comment on this result, and how the DS results in the manuscript might fit (or not!) within the hypothesis of leaf temperature increases (from stomatal closure) driving an increase in isoprene under moderate drought?

*to expand on the context given in the introduction (lines 57-60), it might be worth mentioning how this behaviour has been observed in mature trees in real forests, as described in the references below:

Potosnak, M. J., LeStourgeon, L., Pallardy, S. G., Hosman, K. P., Gu, L., Karl, T., et al. (2014). Observed and modeled ecosystem isoprene fluxes from an oak‐dominated temperate forest and the influence of drought stress. Atmospheric Environment, 84, 314–322. https://doi.org/10.1016/j.atmosenv.2013.11.055

Seco, R., Karl, T., Guenther, A., Hosman, K. P., Pallardy, S. G., Gu, L., et al. (2015). Ecosystem‐scale volatile organic compound fluxes during an extreme drought in a broadleaf temperate forest of the Missouri Ozarks (Central USA). Global Change Biology, 21(10), 3657–3674. https://doi.org/10.1111/gcb.12980 (already in the reference list)

Ferracci, V., Bolas, C. G., Freshwater, R. A., Staniaszek, Z., King, T., Jaars, K., et al., (2020). Continuous Isoprene Measurements in a UK Temperate Forest for a Whole Growing Season: Effects of Drought Stress During the 2018 Heatwave. Geophysical Research Letters, 47(15). https://doi.org/10.1029/2020GL088885

---

## Author Comment (AC1) · 6 Oct 2020

We thank reviewer 1 for the constructive comments. Below is our point by point reply to specific comments.

*Reviewer Comment 1 Line 20: please specify to what levels*
Reply->Line 19: **We exposed plants with daily ozone concentrations of 100 ppb for one hour for seven days, which resulted in faster stomatal closure (e.g. a mean value -31.3% at an average stem water potential of -1 MPa) partially mitigating drought stress effects**.

*Reviewer Comment 2 Line 26: "," should be placed after "Plants" and not after "in".*
Reply: This has been fixed in an updated version of the manuscript.

*Reviewer Comment 3 Line 48 "it has be shown" – please fix.*
Reply: This has been fixed in an updated version of the manuscript.

*Reviewer Comment 4 Line 49: Remove ",".*
Reply: This has been fixed in an updated version of the manuscript.

*Reviewer Comment 5 Line 51: "and longer growing longer seasons" – please rephrase.*
Reply: This has been fixed in an updated version of the manuscript.

*Reviewer Comment 6 Lines 51-54: However, drought can also lead to the opposite effect, as you mention below, which seems to be in conflict with this statement.*
Reply: Here we refer to "enhanced" as an enhancement of the abiotic stress events, not of the BVOC emissions.
Change-> Line 38: **Future climate scenarios with expected temperature increases between 1.8 and 4°C (IPCC, 2007) suggest an additional enhancement of global BVOC emissions between 30 to 45 % (Peñuelas and Lluisiá, 2003). An enhancement of abiotic stress events, due to an indirect effect of a temperature increase (e.g. via ozone or drought episodes) can also alter BVOC emissions (EEA, 2017; Müller et al., 2008; Loreto and Schnitzler, 2010; Dai, 2013; Unger et al., 2013; Sindelarova et al., 2014).**

*Reviewer Comment 7 Line 87 - The hypothesis of this study is based on the research question whether drought and ozone stressors are additives. It would be good to expand more about*
*1) previous study of these two stressors in a broader scope (e.g., (Pollastrini et al., 2014; Wittig et al., 2007)) and*
*2) the importance of currently addressing this specific research question (e.g., how it can help scientists to cope with current assessment/research questions).*
*3) the state of knowledge with respect to BVOC emission (e.g., (Holopainen and Gershenzon, 2010)).*
*Line 87 - "different abiotic stresses" – be specific about whether you refer here only to ozone and drought stressors and/or to additional stressors.*
*Introduction- it will be good to provide information about the role of BVOCs in the troposphere.*
Reply: We added the following clarifications:
1) Line 77: **Pollastrini et al. (2014) consider a change in sensitivity of the plants to ozone (different poplar clones) under severe drought conditions. In their case, ozone and drought produced a synergistic effect for $CO_2$ exchange and chlorophyll fluorescence when applied together. Witting at al. (2007) found a dependency on ozone effects under different levels of drought stress. In fact, Witting et al. (2007) report a dependency of the damage in the photosynthetic apparatus depending on the cumulative ozone flux into the leaf, thus in relation with the stomatal conductance.**

2 and 3) Line 82: In this work, our hypothesis was that **ozone and drought stress** in plants is not necessarily additive, and that the plant's response to drought and ozone exposure can result in an alteration of characteristic BVOC emission strengths. **Changing BVOC emissions have an important impact on climate through atmospheric chemistry (Claeys et al., 2004, Paulot et al., 2009; Hallquist et al., 2009). The presence of BVOCs in atmosphere contribute to the formation of tropospheric ozone and growth of secondary organic aerosol (SOAs), and radicals (Griffin et al., 1999; Orlando et al., 2000; Atkinson and Arey, 2003).**

Line 92: **Understanding how BVOC emissions respond to climate change is therefore essential to understand what direct or indirect effects the biosphere can exert on atmospheric chemistry and climate. A better understanding will also help developing strategies necessary to mitigate the effects of climate change itself (Kulmala et al., 2004; Yuan et al., 2009).**

*Reviewer Comment 8 Lines 93-94 – " its fast-regulated transpiration rates "- This is not clear. I'm not sure that these two factors are sufficient to result in high tolerance to drought.*
Reply: We agree, in particular when comparing with other more drought resistant species. We changed the text as following.
Change->Line 99: According to Ellenberg (1988), **the defensive actions of *Q. robur* against drought stress are caused by fast regulation of transpiration rates and stomatal conductance**, and a low susceptibility of water embolism in the xylem (Van Hees, 1997).

*Reviewer Comment 9 Line 106- "air gas exchange" – Do you mean CO2 and H2O exchange- please be specific in this definition throughout the text (e.g., line 119 and elsewhere).*
Reply: OK, we now add CO2 and H2O where necessary.
Changes: e.g. Line 110: **were used for BVOC emission measurements, CO2 and H2O gas exchange measurements and biochemical** …
Line 132: Throughout the increasing drought stress, tree leaf gas exchange (**$CO_2$ and $H_2O$**) and BVOC emissions were measured for two sets, DS and…
Line 133: Instruments GmbH, Alland, Austria) for 2-3 hours each day in order to measure their **$CO_2$ and $H_2O$ exchange** along with key…
Line 272: Statistics Toolbox Release 2017a, The MatWorks, Inc., Natick, MA, United States). All leaf gas exchange **($CO_2$ and $H_2O$)** and BVOC flux

*Reviewer Comment 10 Can you elaborate on how did you measured this gas exchange?*
Reply: We explain the basic measurement setup on Line 161: $CO_2$ and $H_2O$ mixing ratios in the air leaving the enclosures were measured using a CIRAS-3 SC PP System (Amesbury, MA, USA**), which was factory calibrated three months before the measurement campaign**.

*Reviewer Comment 11 I recommend adding the schematic of the experimental design (Fig. A1) to the Materials and Methods. It will help in following your experiments.*
Reply: This has been fixed in an updated version of the manuscript

*Reviewer Comment 12 Lines 120-121 – please specify if this part still take place in the fitotron.*
Change-> Line 116: **The plants were moved from the greenhouse to an indoor climate chamber** (Fitotron Weiss Gallenkamp, UK) 24h hours before the experiment started. **Thereafter trees were kept in the climate chamber for the remainder of the experiment and were only placed into the branch enclosures during the gas exchange measurements. The branch enclosures were situated next to the climate chamber in a climatized laboratory exhibiting the same environmental conditions as in the climate chamber.** The climate conditions during the first day of experiment were kept at 25°C, ~60 % of **relative humidity (rH)** and ~1000 µmol m$^{-2}$ s$^{-1}$ PAR at canopy top, to

adapt to constant air temperature. To continuously increase the drought stress, the plants were not watered and the humidity in the climate chamber was decreased to 40 % rH and temperature was increased to 30°C after the first day. **The same temperature conditions were also present in the climatized laboratory, where the plants were placed in the enclosures at an rH of 32 % and 30°C. Overall light conditions remained constant during the day, with lights of during the night.**

***Reviewer Comment 12 Line 131- Please define "rH".***
Change-> Line 120: which was kept at 25°C, ~60 % of **relative humidity (rH)** and ~1000 µmol m$^{-2}$ s$^{-1}$ PAR at canopy top, to adapt to constant air temperature.

***Reviewer Comment 13 Line 132- "assured" – can you explain in detail why it assures so?***
Change-> Line 147: The flow rate of 10 l min$^{-1}$, **tested during the experiment set-up prior to the actual experiments**, assured that no…

***Reviewer Comment 14 Lines 135-136- Irradiation took place also in nighttime? Please specify.***
Change-> Line 150: Trees inside the enclosure were LED-irradiated with a mean PAR value of 1374 µmol m$^{-2}$ s$^{-1}$ at canopy top (Eckel Electronics, Trofaiach, Austria) **during daytime when the exchange measurements were performed. During night, trees were kept in the dark**.

***Reviewer Comment 15: Can you provide details about Background (zero) calibration? e.g., if and what frequency and under what conditions such calibration was performed using the PTRTOF6000X2?***
Reply: The PTR-TOF-MS was calibrated daily, this has already been mentioned in the manuscript on line 173. We added some more specificity to describe the background measurements.
Change-> Line 169: The **instrument background was characterized daily during calibrations and in the third empty enclosure that was flushed with background air. Backgrounds were measured every 20 minutes for 5 minutes.**

***Reviewer Comment 16 Line 157 – it is not clear to me what do you mean by "combined calibration uncertainties" and by "a compound specific average experiment sensitivity"***
Change-> Line 174: **Daily measured sensitivities based on compounds in a calibration standard varied on the order of 8-20 % depending on the compound. This lies within the combined calibration uncertainties of the gas standard and dilution setup using two flow controllers. Whenever a compound was not contained in the calibration standard, we applied a compound specific sensitivity using procedures described by Cappelin et al. (2012).**

***Reviewer Comment 16 Line 158 – "40-800 pptv" – it is not clear to me why providing this information is important. It may be more useful to specify specific limits of detection to individual compounds (possibly in Table A3).***
Reply: We take this suggestions and omitted this statement in the revised manuscript.

***Reviewer Comment 17 Lines 161-168 – I suggest to include each individual compound acronym together with its specific m/z (e.g., in parenthesis) and the specific reference, either in the text or in a table.***
Reply: This has been fixed in an updated version of the manuscript

***Reviewer Comment 18 Line 179 – Why don't you include each of the abbreviations in parenthesis?***
Reply: This has been fixed in an updated version of the manuscript

***Reviewer Comment 19 Line 189 – Can you elaborate on how the stomatal resistance was measured? Was it measured for each specific leaf or using another approach?***

Reply: Since the experimental setup consisted of a branch enclosure, stomatal resistance values reflect the bulk average of all leaves enclosed in the enclosure. It was calculated by the application of formula 10.

**Reviewer Comment 20 Lines 189-190 – Can you explain why assuming that the boundary resistance is zero is justified?**

Reply: The high flow rate of 10l/min through our chambers created enough turbulence that the boundary resistance is assumed to be small compared to the stomatal resistance. This can alternatively also be achieved by fans inside the chamber, which we wanted to avoid due to the potential of artifacts from the lubricants that a fan inside the enclosure would cause.

**Reviewer Comment 21 Line 197 – " (see below)" – it would be better to specify the specific section number.**

Reply: This has been fixed in an updated version of the manuscript.

**Reviewer Comment 22 Lines 203-204 – "peroxidase and antioxidant capacity, and phenol content" – it is the first time you mention these properties. It would be good to expand on them and why they were measured.**

Change-> Line 225: **Using foliar materials collected after the seven day period of emission measurements (section 2.2) and stored at -80°C until analysis, peroxidase and antioxidant capacity, and phenol content (TPhe) were measured. These properties provide additional insights in the response of GLV and Shikimate emissions as products of the metabolic process of the enzymatic activity (Betz et al., 2009).**

**Reviewer Comment 22 Line 216 – What do you mean by "linear range of: : :" ?**

Change->Line 241: The activity was calculated **from the slope in the initial linear portion of the reaction progress curved** using an extinction coefficient of $1.13 \times 10^4 \, M^{-1} \, cm^{-1}$ for oxidized *o*-dianisidine (Worthington manual, 1972).

**Reviewer Comment 23 Line 256 – no need for multiple definition in the main text.**

Reply: This has been fixed in an updated version of the manuscript.

**Reviewer Comment 24 Line 260 – " significant." – can you add a P-value?**

Change-> Line 286: R1 and R4 was significant **(p-value 0.02 and 0.05 for DS and DS×OS respectively)**. R1, shown in Fig. 1(a), includes values of trees fumigated with ozone (DS×OS) from the first and

**Reviewer Comment 25 Line 274 - no need for multiple definition in the main text.**

Reply: This has been fixed in an updated version of the manuscript.

**Reviewer Comment 26 Line 285 – " (averagely 96 % of the total emissions" – it would be better to provide this information earlier, so the reader will have this in mind when reading the second paragraph in this section.**

Reply: We moved this portion to Line 303: where we now state: The ratio of CBVOCs and C*A* is shown in Fig. 3. **IS, the dominant BVOC (on average 96 % of the total emissions), mean standardized IS emissions of DS×OS treated plants were consistently higher in all SWP ranges compared to DS alone (Figure 3), thus showing the difference between DS and DS×OS in CBVOCs/C*A* in the highest SWP ratio range**

**Reviewer Comment 27 Line 287 – "carbon loss ratio" - Be more accurate in definition. Do you mean->CIS/CA?**

Reply: This has been fixed in an updated version of the manuscript.

**Reviewer Comment 28 Line 288 –" high drought stress" - Can you specify this in terms of "R"?**

Change->Line 315: At very high drought stress **(R4)** this ratio decreased again to 0.4 in DS and 0.8 in DS×OS.

***Reviewer Comment 29 Line 304 – Please add "." at the end of the sentence.***
Reply: This has been fixed in an updated version of the manuscript

***Reviewer Comment 30 Lines 340-341 – "a decrease of MT emissions" – under what conditions?***
Change-> Line 375: These observations contrast those by Lluisá and Peñuelas (1998) for *Q. coccifera* reporting a decrease of MT emissions **under severe drought conditions**.

***Reviewer Comment 31 Line 366- Is this a new paragraph (if so, make it clear and consistent with the rest of the manuscript)?***
Reply: This has been fixed in an updated version of the manuscript

***Reviewer Comment 32 Lines 385-386 – " by stimulating the phenylpropanoid pathway" – what about the lipoxygenase and hydroperoxide systems?***
Reply: We added a sentence to this section:
Change-> Line 422: On the other hand, DS×OS, showed a small increase of GLV only at the highest stress level. We take this to indicate that ozone has the potential to inhibit drought stress damage and therefore the emissions of GLV, by stimulating the phenylpropanoid pathway to form an antioxidant protection for chloroplasts (Pellegrini et al., 2019**). The GLV emissions in DS×OS are initially inhibited during of the onset of drought. While ozone fumigation initially inhibits the activation of the lipoxygenase and the hydroperoxide lyase pathway indirectly, these pathways are clearly triggered during the progression of severe drought stress (R4) (Heiden et al., 2003; Matzui, 2006).**

***Reviewer Comment 33 Line 396 – " well-watered and severe drought condition" – can you specify these also in terms of R?***
Change->Line 437: The results of our study showed no significant decrease in TPhen due to ozone fumigation both in well-watered and severe drought condition **(R4)** (OS, DS×OS).

***Reviewer Comment 33 Tables 1 and 2 – it is recommended to include the comments below the table.***
Reply: This has been fixed in an updated version of the manuscript.

***Reviewer Comment 33 A1 – Can you include the thermocouple in the figure?***
Reply: We modified the figure and included the thermocouple.

***Reviewer Comment 34 Table A3 – add the compound acronyms/names***
Reply: We added the table in the manuscript.

---

## Author Comment (AC2) · 6 Oct 2020

We thank reviewer 2 for the constructive comments. Below is our point by point reply to specific comments.

*Reviewer Comment 1: The introduction could be more concise and to the point of the hypothesis. I believe there is to much information at first about terpenoid biosynthesis, which if needed could be explained better in detail in the discussions relating it to the results.*
Reply: Thanks for the suggestion; we moved these parts in the results.

*Reviewer Comment 2: When you start talking about temperature as a dominant stress, one asks why you have not used temperature as a treatment, thus removing it (just mention it overall and go directly to the drought and ozone issue).*
Reply: We have incorporated the suggestion with the removal of the two sentences:

*Reviewer Comment 3: The paragraph of the combination of stress is great, but then the biosynthetic information seems misplaced, perhaps do the same as for terpenoid biosynthesis.The objectives must be better explained and put into context in the introduction. Particularly having a paragraph above saying this was already done. Why, having Vitale et al.,2008 and Yuan et al., 2016, we need this study. Additionally, a bit of background about Quercus robur already in the introduction would be interesting, to support why you chose this species (more than a widely distributed isoprene emitting oak species,i.e. is this species going to suffer drought and ozone stress in particular? Why?).*
Reply: Now we rearranged the introduction, for a better understanding.
Change ->Line 69: Few studies have analyzed the effects of plant emissions from a combination of drought and ozone stress (Vitale et al., 2008; Yuan et al., 2016). Studying *Quercus ilex*, Vitale et al. (2008) reported that drought stress leads to stomatal closure therefore reducing stress by ozone as it is restricted to enter the leaf. **They did not report effects of ozone when going from a well watered situation to severe stress.** Yuan et al. (2016) found that drought increased isoprene emissions in a hybrid poplar deltoid species, but that isoprene emissions decreased under moderate drought stress combined with long-term ozone fumigation. **. In their case, Yuan et al. (2016) analyzed the emissions in a situation of moderate drought stress.**

**Here we are also interested in the situation of severe stress that could occur in the future due to climate change, combined with model projections of elevated ozone concentrations (> 100 ppb).**

**Pollastrini et al. (2014) consider a change in sensitivity of the plants to ozone (different poplar clones) under severe drought conditions. In their case, ozone and drought produced a synergistic effect for $CO_2$ exchange and chlorophyll fluorescence when applied together. Witting at al. (2007) found a dependency on ozone effects under different levels of drought stress. In fact, Witting et al. (2007) report a dependency of the damage in the photosynthetic apparatus depending on the cumulative ozone flux into the leaf, thus in relation with the stomatal conductance.**

In this work, our hypothesis was that **ozone and drought stress** in plants is not necessarily additive, and that the plant's response to drought and ozone exposure can result in an alteration of characteristic BVOC emission strengths. **Changing BVOC emissions have an important impact on climate through atmospheric chemistry**

**(Claeys et al., 2004, Paulot et al., 2009; Hallquist et al., 2009). The presence of BVOCs in atmosphere contribute to the formation of tropospheric ozone and growth of secondary organic aerosol (SOAs), and radicals (Griffin et al., 1999; Orlando et al., 2000; Atkinson and Arey, 2003).**

As a model plant we chose *Quercus robur* L., a widely distributed isoprene emitting oak species in Europe (Barstow and Khela, 2017**), considered not at risk of extinction (Barstow and Khela, 2017). In the future, this species may become more threatened (Barstow and Khela, 2017), triggering a migration from the current climate range to a zone more representative of the north and east of Europe (EFDAC, 2015). Climate change could also expose *Q. robur* to greater environmental stress from drought (Jonsson, 2012). Understanding how BVOC emissions respond to climate change is therefore essential to understand what direct or indirect actions they can have on the biosphere-atmosphere-climate system and to develop strategies necessary to mitigate the effects of climate change itself (Kulmala et al., 2004; Yuan et al., 2009).**

*Reviewer Comment 4: Methodology must be better explained. In particular a diagram choosing the number of replicates chosen for each treatment. You say you have 14 trees in total, how are they separated. For instance R4 only has two replicates for DSxOS, why? Additionally a time series of watering and lack of watering could be expressed in this diagram as well. It is confusing what you use for emission measurements, for biochemical assays and for references. To sum up the methodology must be revised and better explained. Think that the reader must be able to reproduce your methodology. More detailed to be found below. Line 106: I am really concern with plant acclimation here. As far as I understand the plants are moved ONLY 24 hours before measurements to the climate chamber. Is this enough? Please argue how is this enough. What do you mean by to adapt to constant air temp?*

Reply: Rather than including an additional diagram we decided to improve the description of the methodology where necessary in the text to make it more clear. We addressed other important comments in the revised paper. Briefly, R4 (like other groups) is grouped such that it represents a specific stress level in SWP by the plants. The replicates for DSxOS were envisioned to be at least 3 for all experiments, however one replicate of this particular set (R4) did not reach the required level of stress at the end of the experiment and had therefore be associated with R3 instead. Generally we acknowledge the reviewers comment that more replicates would always be better, but this is often limited by the experimental capability. Generally 3 true replicates were envisioned for these experiments. In addition by using branch enclosures, rather than sampling individual leaves, an experimental average of many individuals for each treatment was obtained, minimizing leaf to leaf variability. Prior to experiments plants were kept in a greenhouse outside the laboratory exhibiting environmental conditions (daily average T: 22.5 +/3 °C and RH: 54%) comparable with the conditions in the phytotron (25 +/-2 °C, RH 60%) and subsequently branch enclosures.  Plants were moved to the phytotron 24h prior to the experiments and thereafter housed under exactly the same conditions between the branch enclosure setup and the phytotron. Due to small changes between the greenhouse and laboratory experiements we do believe 24h acclimation was sufficient. This is also corroborated  by well established BVOC emission algorithms (Guenther et al., 1999) showing that the 24h period is the most important one for acclimation, with the previous 240h playing a comparably smaller influence. In our case the impact on isoprene emissions for a scenario of 23 °C 240h temperature history rather than 25 °C would be for example con the order of 4-5%.

Change-> Line 108: For the biochemical **reference** assays**,** eight trees **of the initial fourteen were used:** four well-watered plants (C) and four well-watered plants receiving one time 100 ppb ozone for one hour (OS) inside the enclosures. **The remainder (six plants) were used for BVOC emission measurements, CO2 and H2O gas exchange measurements and biochemical assays. Hereby we were mostly left with three replicates under drought stress (DS) and three replicates exposed to drought stress and ozone (DSxOS). The drought stress was initiated, for all the six plants, 10 days** before the VOC measurements started and was maintained by keeping the soil water content at 4-5 vol.% using a soil moisture probe (Fieldscout TDR100, 20 cm probe depth, Spectrum 105 Technologies, UK), whereas 100 % field capacity was 13.4 vol.%**. Starting the VOC measurements, we stopped watering the previously drought stressed trees to increase further drought stress**.

***Reviewer Comment 5: Line 96: where do the 2-year-old trees are coming from?***

Reply: The trees are from the tree school Natlacen in Pilgersdorf, in the south-east of Austria. The city gardeners of Vienna (MA42) are ordering their trees from the same tree school for replanting or newly planting street trees. Usually these trees are a couple of years older than the ones we received from them, but since our VOC-chambers are too small, we were able to get a hold of the old 2-year old saplings.

***Reviewer Comment 6: Line 97: What do you mean by soil used by the city gardeners? What brand?***

Reply: The MA42 (Magistrate no. 42) is responsible for Viennese park and city vegetation. Together with the ÖGLA (Österreichische Gesellschaft für Landschaftsarichtektur) they developed a customized soil mixture, which holds the water for a longer time to prevent early drought stress during long dry periods. Further information about the soil can be found on the webpage http://oegla.at/uebersicht/125-seminarrreihe-baum-download-unterlagen - "Das Wiener Baumsubstrat" (Fluvial fine sediment of the Danube, compost, sand and dolomite grit).

***Reviewer Comment 7: Line 97: What brand is the quartz sand from?***

Reply: We used filter sand (purchased from Obi, article no. 6270599) with a grain size between 0.7-1.2mm fulfilling the criteria of DIN EN 15798 (used for filtering swimming pool water).

***Reviewer Comment 8: Line 98: how much fertilizer you put?***

Reply: We used the recommended amounts for small trees: 5 caps fertilizer mixed in 10L water for 4m².

***Reviewer Comment 9: Line 99: Tulln is a place not a brand...put the brand of the greenhouse or say how did you reach the levels mentioned.***

Reply: Yes we acknowledge your comment and changed the text accordingly:

Change->Line 103: The plants were fertilized once after planting (universal fertilizer NovaTec, Compo, Münster, Germany) and from thereon kept well-watered in a greenhouse at near ambient light **(80 % to 90 % of photosynthetically active radiation) (Lak et al., 2020)**.

***Reviewer Comment 10: Line 100: what do you mean by close by experiment Line 101: Please state better the time of measurements? The biochemical assays should also be explained in the diagram Line 104: where do you perform the drought stress, in what conditions are the plants? Line 105: I do not understand what do you mean by "maintained by keeping the soil water content at 4-5 vol%". Wasn't this a continuously increasing drought? Was this maintained at all SWP ranges? Then the control plants were at field capacity? Please explain better***

Reply: Trees were moved to the greenhouse in Vienna on July 5th 2020. The experiment started in Vienna on July 15th 2019. We changed the original text regarding the biochemical assays rather than including an additional diagram. Specific changes requested by the reviewer are now summarized as following:

Change-> Line 105: The trees were moved **from a greenhouse in Tulln** into another close-by greenhouse in Vienna two weeks prior to the experiments. Dust was removed from the leaves by showering the trees before starting the drought stress**.**

For the biochemical **reference** assays**,** eight trees **of the initial fourteen were used:** four well-watered plants (C) and four well-watered plants receiving one time 100 ppb ozone for one hour (OS) inside the enclosures. **The remainder (six plants) were used for BVOC emission measurements, CO2 and H2O gas exchange measurements and biochemical assays. Hereby we were mostly left with three replicates under drought stress (DS) and three replicates exposed to drought stress and ozone (DSxOS). The drought stress was initiated, for all the six plants, 10 days before the VOC measurements started and was maintained by keeping the soil water content at 4-5 vol.% using a soil moisture probe (Fieldscout TDR100, 20 cm probe depth, Spectrum 105 Technologies, UK), whereas 100 % field capacity was 13.4 vol.%.** Starting the VOC measurements, we stopped watering the previously drought stressed trees to increase further drought stress.

***Reviewer Comment 11: Line 108: what do you mean by mid canopy height?***
Reply: We measured the height of the plants and the conditions inside the climate chamber at the mid canopy height.

***Reviewer Comment 12: What was the PAR level at the climate chamber?***
Reply: The value was ~1000 µmol m$^{-2}$ s$^{-1}$ PAR at canopy top.

***Reviewer Comment 13: Line 111:So DSxOS individuals are fumigated with ozone inside the enclosure while measurements were taken place or prior measurements? Please state.***
Change->Line 127: two groups, three trees were drought stressed and fumigated with 100 ppb O3 (DS×OS) inside the enclosure for one hour each day after **the daily** measurement of BVOCs.

***Reviewer Comment 14: Line 113: why humidity was decrease and temperature increased to maintain the drought strees? Wasn't this maintained by the SWC?***
Reply: Line 121: To continuously increase the drought stress, the plants were not watered and the humidity in the climate chamber was decreased to 40 % rH and temperature was increased to 30°C after the first day. The same temperature conditions were also present in the climatized laboratory, where the plants were placed in the enclosures at a rH of 32 % and 30°C.-> we changed the humidity and the temperature in the climate chamber for increase the drought stress, don't water the plants was not enough to increase the drought stress, so we decreased the humidity and increased the temperature.

***Reviewer Comment 15: Line 115: what is C and what is OS?***
Reply: C were control plants (well-watered), OS were the well-watered plans plus an ozone fumigation of 100 ppb for one hour. This is now more explicitly explained throughout the text

***Reviewer Comment 16: Additionally do the plants stay there or they go back to the greenhouse? I hope they stay in the climate chamber, otherwise it wouldn't be right. Please state. Line 119: what do you mean by tree leaf gas exchange? Please state what do you mean by gas exchange and also why not this is tree emission as the branch is also inside the cuvette. Line121: as far as I understand you maintain the tree during the seven days inside the cuvette? Do you have as many cuvettes or only 4? Please explain better***
Reply: We did not use leaf cuvettes, but whole plant enclosures instead, to minimize leaf to leaf variability in these experiments. Due to the flow demand and experimental design we were limited to 4 branch enclosures. The trees were first moved to the climate chamber 24h prior to the start of experiments. The climate chambers were housed inside a climatized laboratory, where measurements took place. The chambers were set up such that the climate conditions in the climate chamber (T, RH, PAR, CO$_2$) matched conditions in the laboratory where the experiments took place. The reason for this setup was that the climate chambers themselves were too small to house the entire experimental setup. During the drought experiment two sets with 3 replicates were measured in the branch enclosures daily. At the beginning of each day the trees of the first set were placed in the 3 branch enclosures and continuously monitored for 2h. Readings of the last 20minutes from these 2 hours were then averaged for further analysis. We assured that VOC profiles were in steady state after placing trees in the branch enclosures and verified this by continuously monitoring BVOC concentrations and gas exchange inside the bags for at least 2h. After the first set was measured, trees were placed back in the climate chamber and the second set of trees was put in the branch enclosures. Overall, trees were kept 3h in the branch enclosure each day on average. For the rest of the day they remained in the climate chamber.
Change-> Line 116: **The plants were moved from the greenhouse to an indoor climate chamber** (Fitotron Weiss Gallenkamp, UK) 24h hours before the experiment started. **Thereafter trees were kept in the climate chamber for the remainder of the experiment and were only placed into the branch enclosures during the gas exchange measurements. The branch enclosures were situated next to the climate chamber in a climatized laboratory exhibiting the same environmental conditions as in the climate chamber.** The climate conditions during the first

day of experiment were kept at 25°C, ~60 % of **relative humidity (rH)** and ~1000 µmol m-² s-¹ PAR at canopy top, to adapt to constant air temperature. To continuously increase the drought stress, the plants were not watered and the humidity in the climate chamber was decreased to 40 % rH and temperature was increased to 30°C after the first day. The same temperature conditions were also present in the climatized laboratory, where the plants were placed in the enclosures at a rH of 32 % and 30°C. **Overall light conditions remained constant during the day, with lights of during the night.**

Change->Line 134: The plants were taken out of the climate chamber and kept inside **the** custom-made plant enclosures (**Fig.** 1; TC-400, Vienna Scientific Instruments GmbH, Alland, Austria) for 2-3 hours each day in order to measure their **$CO_2$ and $H_2O$ exchange** along with key physiological parameters (soil moisture and stem water potential). **After the measurements inside the enclosures, the plants were taken back to the climate chamber until the next measurement session.**

***Reviewer Comment 17: Line 131: why only 370 ppm of CO2, is this realistic to nowadays?***
Reply: We used ambient $CO_2$ concentrations in our experiments that were present in the laboratory and climate chamber as well as outside during this season. So we believe this value is within the current variability on the ground, but acknowledge that annual concentrations are nowadays typically 8 % higher.

***Reviewer Comment 18: Line 145: how did you calibrate for CO2 and H2O?***
Reply: Thanks for making this point clear, we changed as follows:
Change->Line 161: $CO_2$ and $H_2O$ mixing ratios in the air leaving the enclosures were measured using a CIRAS-3 SC PP System (Amesbury, MA, USA**), which was factory calibrated three months before the measurement campaign.**

***Reviewer Comment 19: Line 155: please state the compounds inside***.
Reply: We added Table A3 in the appendix with the compounds used for the calibration.
Change-> Line 172: containing 15 compounds **(Table A3)** with different functionality distributed over a mass range of 33-137 amu were performed daily.

Table A3: m/z ratio and chemical formula and name of compounds presents in the standard gas mixture used for the daily calibration of the PTR-Tof-MS.

| m/z ratio | Chemical formula | Compound |
|---|---|---|
| 32.0262 | $CH_3OH$ | Methanol |
| 41.0265 | $C_2H_3N$ | Acetonitrile |
| 44.0261 | $C_2H_4O$ | Acetaldehyde |
| 58.0418 | $C_3H_6O$ | Acetone |
| 72.0574 | $C_4H_8O$ | Methyl Ethyl Ketone (MEK) |
| 78.0469 | $C_6H_6$ | Benzene |
| 92.0625 | $C_7H_8$ | Toluene |
| 106.0782 | $C_8H_{10}$ | Xylenes |
| 120.0939 | $C_9H_{12}$ | 1,2,4-Trimethylbenzene (TMB) |
| 136.1252 | $C_{10}H_{16}$ | a-Pinene |
| 62.0189 | $C_2H_6S$ | Dimethyl sulphide (DMS) |

| | | |
|---|---|---|
| 86.0731 | $C_5H_{10}O$ | 2-methyl-3-buten-2-ol (MBO) |
| 134.1095 | $C_{10}H_{14}$ | 1,2,4,5-Tetramethylbenzene |

**Reviewer Comment 20: Line 155: why did you have to perform calibrations so often?**

Reply: We performed these calibrations often enough to assure good experimental results from the PTR-qiTOFMS during the experiment. Since we used a brand new instrument for these experiments we also wanted to assure that the performance of the instrument was adequate.

**Reviewer Comment 21: Line 152: do you use an average calibration factor for all compounds? Which is certainly not correct but at least what I interpret from the text. Please state how do you specifically calibrate for GLV, MESA and Sqt. Do you have them in your calibration bottle?**

Reply: When we have the compound in the gas standard we use an average value of all calibrations for that specific compound. For not directly calibrated compounds (including GLV & MESA) we extrapolated the sensitivity of measured compounds according to procedures described by Cappellin et al. 2012: (doi: 10.1021/es203985t ).

**Reviewer Comment 22 Line 161-166: please talk about possible contributors to this mass...how are you sure you can only attribute those signals to the mentioned compounds?**

Reply: For isoprene and monoterpenes the uniqueness was verified by a set of parallel measurements using a GC-MS sampling setup (Fitzky et al. in prep.2020). For other compounds there is a wide body of literature of likely candidates that have been identified over the past decades as cited. Using PTR-TOF-MS we can obtain an actual isobaric formula, eliminating a range of potentially interfering species compared to older technology (e.g. QMS). Yet it is true that potential interferences are always possible with in-situ instrumentation. So strictly speaking our results refer to the isobaric formulas which are now cited throughout the manuscript.

Changes: We are now more specific about the suggested species assignment and refer to the actual isobaric formula in the first place and mention likely VOCs contributing to the individual isobaric formulas.

**Reviewer Comment 23: Line 193: please name and comment on the calibration of these compounds.**

Reply: The ratio of the sum of carbon lost in form of BVOC ($C_{BVOCs}$) vs. the uptake of carbon from net photosynthesis ($C_A$) was calculated according to Pegoraro et al. (2004), with the BVOCs used to calculate $C_{BVOCs}$ given in Table A2 (Line 215).

The list of the compounds is summarized in Table A2. For the calibration of these compounds we used the calibration gas used for the calibration of the PTR.

**Reviewer Comment 24: Line 254: why high to moderate, any references?**

Change-> Line 280: All six trees began the experiment with a high to moderate mean SWP of -0.9 MPa **(Brüggemann and Schnitzler, 2002)**

**Reviewer Comment 25: 260-263: please rephrase, I just don't get it.**

Reply: The grouping performed for this analysis was based on SWP, and not time or day, because it reflects the actual physiological changes. We therefore clarified this paragraph as following.

Change-> Line 286: R1, shown in Fig. 2 (a), includes values of trees fumigated with ozone (DS×OS) from the first and the second day of analysis, **because, for this set, SWP hadn't changed much during these two days.** Differently**, for DS,** R1 includes only measurements of the first day. **The values collected during the second day of analysis, for the set DS, is assigned to R2, because we observed a decreased of SWP between the first and second day of measurement.**

**Reviewer Comment 26: Line 303: wouldn't it be better to say R4 instead of SWP -6MPa?**

Change-> Line 332: emissions at **R4**.

***Reviewer Comment 27: Line 310: please can you mention on how they did not change?***

Change->Line 340: In contrast**, no significant increase was observed in the leaf temperatures,** suggesting IS emissions of DS×OS in R2 being a result of a temperature-independent isoprene production.

***Reviewer Comment 28: Line 334: actually for MT DS there was no significant increase.....***

Reply: Is significant the increase between R1 and R4 for DS*.*

Change-> Line 373: In this experiment MT emissions from *Q. robur* increased in DS and DS×OS trees. In the case of DS, there was a positive effect of drought, with **a significant** increase in MT emissions, although there was a drastic decrease of IS emissions when the water deficit was severe.

***Reviewer Comment 29: Line373-374: please state the values of low and high***

Change-> Line 409: In this experiment, GLV emissions in R4 **were not significantly different from R1, with low values** in ozone treated plants (DS×OS), while plants that were exposed to drought only (DS) exhibited higher emissions, **with a significant increase of GVL emissions between R1 and R4 (Table 2).**

***Reviewer Comment 30: Line 404: please do not use the word believe!***

Change-> Line 444: We **consider** that leaves

---

## Author Comment (AC3) · 6 Oct 2020

Dear Valerio Ferracci, we appreciate your comment.

In our experiment, isoprene emissions are standardized for temperature (30 °C), and we do not expect a large variation caused by short-term changes in temperature. The temperature effect observed in ecosystem scale studies is therefore largely normalized for in our experiments. In field campaigns, isoprene emissions are often observed to stay high during progressive drought and this as you point out has often been ascribed to a compensation by increasing leaf temperatures during drought.

[Figure]

We also note that leaf temperatures remained quite constant in our experiments despite stomata closing. Our interpretation of this is the vigorous mixing of air in the chamber (and thus the boundary layer conductance going to infinity) so that most of the energy is exchanged as sensible heat and stomatal closure does not affect leaf temperature a lot. This could be the explanation for lack of elevated emissions and the observed continuous decrease in isoprene emission. Under field conditions, as in the three cited manuscripts this might be different.

Arianna Peron and Thomas Karl